# Aequa: Fair Model Rewards in Collaborative Learning via Slimmable Networks

**Nurbek Tastan**[1]  **Samuel Horváth**[1]  **Karthik Nandakumar**[1,2]

## Abstract

Collaborative learning enables multiple participants to learn a single global model by exchanging focused updates instead of sharing data. One of the core challenges in collaborative learning is ensuring that participants are rewarded fairly for their contributions, which entails two key sub-problems: *contribution assessment* and *reward allocation*. This work focuses on fair reward allocation, where the participants are incentivized through *model rewards* - differentiated final models whose performance is commensurate with the contribution. In this work, we leverage the concept of **slimmable neural networks** to collaboratively learn a shared global model whose performance degrades gracefully with a reduction in model width. We also propose a post-training **fair allocation algorithm** that determines the model width for each participant based on their contributions. We theoretically study the convergence of our proposed approach and empirically validate it using extensive experiments on different datasets and architectures. We also extend our approach to enable training-time model reward allocation. The code can be found at https://github.com/tnurbek/aequa.

## 1. Introduction

Collaborative learning (CL) has emerged as a transformative paradigm for training machine learning models across data silos while preserving data privacy. Unlike centralized approaches, CL enables participants (e.g., hospitals, financial institutions, etc.) to jointly train a shared global model by exchanging only focused updates rather than raw data (McMahan et al., 2017). CL mitigates privacy risks and complies with regulations such as GDPR, making it particularly

[1]Mohamed bin Zayed University of Artificial Intelligence (MBZUAI), Abu Dhabi, UAE [2]Michigan State University (MSU), Michigan, USA. Correspondence to: Nurbek Tastan <nurbek.tastan@mbzuai.ac.ae>.

*Proceedings of the 42nd International Conference on Machine Learning*, Vancouver, Canada. PMLR 267, 2025. Copyright 2025 by the author(s).

useful in domains where data sensitivity is paramount. A special case of CL is federated learning (FL), where a central server orchestrates the collaboration. However, CL/FL faces many challenges such as communication inefficiencies due to frequent exchange of updates, system heterogeneity resulting from different participant capabilities, and statistical heterogeneity caused by non-i.i.d. data distributions across participants (Zhu et al., 2021). These issues often degrade model performance, scalability, and practical adoption.

A critical yet less explored challenge in CL lies in ensuring **collaborative fairness** among participants. Traditional CL frameworks assume uniform contributions from all parties, but real-world scenarios involve disparities in data quality, quantity, and computational resources. For instance, participants with high-quality data may receive disproportionately less rewards despite their critical role in model generalization. The interplay between fairness and incentivization is essential to sustaining long-term collaboration. Without equitable incentives, participants may withhold resources or disengage entirely, leading to the "free-rider problem" where some entities benefit without contributing meaningfully. To address this issue, recent research has explored incentivization mechanisms such as Shapley value-based reward allocation (Xu et al., 2021; Tastan et al., 2024a), reputation systems (Xu & Lyu, 2020), and game-theoretic frameworks (Wu et al., 2024). These methods aim to quantify and reward participant contributions transparently.

Achieving collaborative fairness in CL fundamentally hinges on two key sub-problems: (1) contribution assessment and (2) reward allocation mechanism. Contribution assessment evaluates the marginal impact (contribution) of each participant's data or computational resources on the overall performance of the global model. Recent efforts (Xu et al., 2021; Jia et al., 2019; Shi et al., 2022; Jiang et al., 2023) attempt to quantify the quality of the local model updates as a proxy for measuring their actual contribution. Once the marginal contribution of each participant is determined, a reward allocation mechanism is necessary for incentivizing the participants to collaborate. Rewards can be in the form of financial compensation (monetary rewards) or differentiated final models (model rewards). To ensure fairness, rewards must be commensurate with the contribution. **This work focuses exclusively on fair distribution of model rewards in CL**.

A critical question arises: *How can participants be rewarded with different models whose performance (accuracy) faithfully reflects their heterogeneous contributions?* While previous work has explored this problem (Xu et al., 2021; Wu et al., 2024), they typically rely on sharing partial updates with participants, which often lacks rigorous convergence guarantees. While (Wu et al., 2024) provide convergence analysis, their approach suffers from the following limitations: (1) The introduction of a stochastic recovery mechanism necessitates occasional broadcasting of the full global model to all participants, thereby aiding free-riding; (2) Since participants begin training from divergent starting points in each round (except the initial broadcast), it leads to potential instability in model convergence; (3) Finally, sampling a subset of updates from a pool of gradients does not inherently ensure that low-contribution participants receive low-quality models; they may still receive high-quality gradients originating from high-contribution participants.

To circumvent these problems, we draw inspiration from the concept of slimmable networks (Yu et al., 2019; Yu & Huang, 2019b), which were originally proposed to dynamically adjust the width of neural network models for efficient inference with a graceful degradation in model performance. By extending this concept to the collaborative/federated setting, we obtain a global model with a nested structure, where subnetworks of varying widths (e.g. $0.25\times, 0.5\times, 1.0\times$) are embedded. Participants are then assigned subnetworks corresponding to their contribution levels – higher contributors receive wider, higher-performing subnetworks, while lower contributors obtain narrower ones. Our approach ensures that model rewards are proportional to client contributions, achieving both high performance and collaborative fairness simultaneously. However, there is one significant obstacle that needs to be surmounted. Collaborative learning in the plaintext domain exposes the intermediate models to the participants, thereby re-introducing the free-rider problem. To overcome this limitation, we assume that each participant has access to a trusted execution environment (TEE), and local training happens confidentially within a TEE. While the use of TEEs has been considered in the FL literature (Huba et al., 2022; Eichner et al., 2024; Daly et al., 2024), they are most used on the server side for secure aggregation and minimization of privacy risks. This work uses TEEs on the client side to enhance collaborative fairness.

The main contributions of this work are as follows: (1) We introduce a CL framework called **Aequa** (*Latin: fair*) that leverages slimmable networks to dynamically adapt model performance to client contributions, serving as a reward mechanism that complements any contribution assessment method. (2) We propose a fair allocation mechanism for post-training distribution of model rewards and then extend this approach for training-time rewards. (3) We provide convergence analysis demonstrating that our framework retains optimality guarantees. (4) We also empirically validate the efficacy of our framework through experiments on benchmark datasets, highlighting balanced model performance and fairness across diverse scenarios.

## 2. Related Work

**Fairness in FL.** Fairness in federated learning has been extensively studied through two primary lenses: performance fairness (Jiang et al., 2023), which emphasizes uniform model performance across all participants, often via personalization strategies such as (Li et al., 2021) that learn a client-specific model regularized toward a shared global model, and collaborative fairness (Lyu et al., 2020), which advocates proportionality between client contributions and rewards. Our work focuses on collaborative fairness, where clients receive model rewards commensurate with their contributions. The foundational work of Lyu et al. (2020) operationalizes collaborative fairness by assigning only the allocated aggregated updates based on their reputations. Other studies consider fairness by quantifying the impact of clients on the global model – the naive choice being the self-reported dataset sizes (self-reported information) (Donahue & Kleinberg, 2021; Zhang et al., 2020), and similarly, Kang et al. (2019) employ such self-reported information to build a fair scheme based on contract theory. Various approaches also assess client importance through Shapley values (Shapley, 1953; Ghorbani & Zou, 2019), utility games (Gollapudi et al., 2017; Nishio et al., 2020) and empirical methods (Shyn et al., 2021). For a complete taxonomy of fairness in FL, we refer the reader to check (Shi et al., 2024).

**Contribution assessment.** A substantial body of work has addressed the problem of evaluating individual client contributions in federated learning. As mentioned earlier, an initial approach to collaborative fairness (Lyu et al., 2020) employed a global validation set, applying a function $\sinh$ to the validation accuracy of each client as a penalty mechanism to approximate their contribution or reputation. Subsequently, Xu et al. (2021) removed the need for a global validation set by approximating game-theoretic Shapley values with the cosine similarity of shared parameter updates – thereby capturing each client's marginal contribution. A range of follow-up studies (Shi et al., 2022; Jiang et al., 2023; Lin et al., 2023; Tastan et al., 2024a) further expanded and refined these strategies for contribution assessment.

**Reward mechanisms.** Broadly, existing incentive mechanisms in FL fall into two categories: post-training monetary rewards and training time model rewards. The former employs frameworks such as Stackleberg games (Zhan et al., 2020), auctions (Zhang et al., 2021; Cong et al., 2020), and contract theory (Liu et al., 2022a; Yang et al., 2024) to distribute monetary compensation post hoc based on client contributions. The latter focuses on model-based rewards

during training, incentivizing participation by dynamically adjusting access to the model's capabilities. For example, CGSV (Xu et al., 2021) allocates sparsified model outputs to clients proportionate to their contributions, while achieving fairness. Similarly, IAFL (Wu et al., 2024) shares aggregated gradients based on each client's contribution through probabilistic sampling, thus restricting highly performing models from under-contributing clients. Another related approach is proposed by Lin et al. (2023), who aim to balance collaborative fairness with long-term equality by introducing an explore-then-exploit mechanism that estimates client contributions over time and gradually allocates rewards to ensure asymptotic parity across clients. While CGSV relies on a heuristic approach and lacks a formal convergence analysis, IAFL includes a convergence proof but exhibits its own limitations. Specifically, its stochastic recovery mechanism shares the full model updates with all participants based on a certain probability: setting this probability to zero yields higher fairness at the expense of performance, and increasing it boosts performance at the cost of fairness – yet still falls short of the performance achieved by FedAvg (McMahan et al., 2017).

**Slimmable networks.** The seminal work by (Yu et al., 2019) introduced the idea of training a single neural network that can operate at multiple widths, enabling dynamic trade-offs between model size and performance. This innovation led to numerous follow-up studies and applications in federated learning, predominantly focused on resource efficiency (Mei et al., 2022; Horváth et al., 2021), communication and computational efficiency (Wang et al., 2022), and neural architecture search (Yu & Huang, 2019a). To the best of our knowledge, we are the first to leverage slimmable networks in the context of fair federated learning.

## 3. Preliminaries

We consider a federated learning (FL) setup with $N$ participants collaboratively learning the parameters $\boldsymbol{x} \in \mathbb{R}^d$ of a machine learning model. Each client $i$ possesses a local dataset $\mathcal{D}_i$ and the overall objective is to minimize a sum-structured FL optimization problem, given by

$$\boldsymbol{x}^* \leftarrow \arg\min_{\boldsymbol{x} \in \mathbb{R}^d} \left[ F(\boldsymbol{x}) := \frac{1}{N} \sum_{i=1}^N F_i(\boldsymbol{x}) \right], \quad (1)$$

where the local loss components $F_i : \mathbb{R}^d \to \mathbb{R}$ are distributed among $N$ participants and are expressed in a stochastic format as $F_i(\boldsymbol{x}) := \mathbb{E}_{\xi \sim \mathcal{D}_i}[F_i(\boldsymbol{x}, \xi)]$.

Our specific task is supervised classification, where we define a classifier $\mathcal{C}_{\boldsymbol{x}}$ parameterized by $\boldsymbol{x}$, mapping input samples to class labels $\mathcal{C}_{\boldsymbol{x}} : \mathcal{Z} \to \mathcal{Y}$, where $\mathcal{Z} \subseteq \mathbb{R}^D$ represents the input space, $\mathcal{Y} = \{1, 2, \ldots, C\}$ denotes the label space, $D$ is the input dimensionality, and $C$ is the

---

**Algorithm 1** Aequa: Federated optimization

**Input:** minimum width $p_{\min}$, maximum width $p_{\max}$, comm. rounds $T$, number of participants $N$, randomly initialized parameters $\boldsymbol{x}^0$, number of local iterations $E$
1: **for** each round $t \leftarrow 0, 1, \ldots, T-1$ **do**
2:     Server broadcasts $\boldsymbol{x}^t$ to each client $i, \forall i \in [N]$
3:     **for** each participant $i \in [N]$ **do**
4:         **for** $k \leftarrow 0, 1, \ldots, E$ **do** ▷ *Local iterations*
5:             Sample width $p_{(i,k)} \leftarrow \mathcal{U}([p_{\min}, p_{\max}])$, where $\mathcal{U}([a, b])$ represents uniform distribution in $[a, b]$.
6:             Update parameters of the model corresponding to $p_{\max}$ and $p_{(i,k)}$
7:         **end for**
8:         Send the updated parameters $\boldsymbol{x}^t_{(i,E)}$ to the server
9:     **end for**
10:    Server updates $\boldsymbol{x}^{t+1} \leftarrow \frac{1}{N} \sum_{i=1}^N \boldsymbol{x}^t_{(i,E)}$
11: **end for**

---

number of classes. The empirical loss function for each client is defined as $F_i(\boldsymbol{x}, \xi) = \mathcal{L}(\mathcal{C}_{\boldsymbol{x}}(z), y)$ where $\mathcal{L}$ is the loss function and $\xi := (z, y)$ is a training sample drawn from the local dataset of participant $i$. Additionally, the model parameters $\boldsymbol{x}$ must be *slimmable*, i.e., it should be possible to obtain a subnetwork $\boldsymbol{x}(p) \subseteq \boldsymbol{x}$ by dynamically adjusting the width parameter $p$, such that the model performance is proportional to the width (which in turn is set proportional to the participant contribution).

## 4. Proposed Solution

In federated learning, ensuring fairness in model allocation is a fundamental challenge, as clients contribute to training with varying levels of data quality, quantity, and computational resources. To address this, we propose an allocation mechanism based on slimmable networks, which ensures fair model rewards by adjusting the model width assigned to each client commensurate to their contributions. This section details our approach to federated optimization (Section 4.1), the allocation algorithm (Section 4.2) and the extension to training-time model rewards (Section 4.3).

### 4.1. Federated Optimization

Slimmable networks are a class of deep learning architectures that allow dynamic adjustments of model width, ensuring that different clients can operate with models of varying capacities without training separate networks. Initially introduced for efficient model scaling (Rippel et al., 2014; Yu et al., 2019; Yu & Huang, 2019b; Horváth et al., 2021; Kusupati et al., 2022; Horváth et al., 2023), we repurpose slimmable networks for fairness in federated learning, ensuring that model allocation reflects each client's contribution to training. Each slimmable network can switch between

different widths $p \in [p_{\min}, p_{\max}]$, where $p_{\max}$ represents the full model and $p_{\min}$ is the smallest subnetwork. Intermediate widths allow for a smooth transition based on contribution levels.

Incorporating slimmable networks into FL requires adapting the optimization process to ensure that all subnetworks contribute effectively to the learning process. In our approach, clients train with varying model widths at each local iteration, ensuring that all width configurations are updated. The training process follows a random-width sampling strategy, where each client trains on different subnetworks in each iteration (one forward-backward pass), promoting a balanced optimization process across all model sizes.

The federated optimization process begins with the server initializing the global model and setting the minimum and maximum possible widths. In each communication round, the server broadcasts the full model to all clients, who then train locally using a uniformly sampled width from the allowed range $[p_{\min}, p_{\max}]$. Clients update both the full model parameters and the parameters corresponding to their sampled width, following a strategy similar to the sandwich rule in (Yu & Huang, 2019b), but with improved efficiency. Upon completing local training, clients transmit their updated weights to the server, which then aggregates them to update the global model.

### 4.2. Fair Allocation Algorithm

**The allocation problem.** We consider a federated learning setting with $N$ clients, indexed by $i = 1, 2, \ldots, N$. Each client has an associated contribution score, represented by the vector $\boldsymbol{c} = (c_1, c_2, \ldots, c_N)$, where $c_i$ denotes the contribution (e.g., standalone accuracy as a surrogate) of the client $i$. Without loss of generality, we assume that the clients are sorted such that $c_1 \leq c_2 \leq \cdots \leq c_N$. We have access to a (sufficiently large) family of models whose performances $\{a_k\}$ lie within a continuous interval $[\ell, u]$, where $u \geq c_N$ represents the highest achievable accuracy and $\ell \leq c_1 + (u - c_N)$ denotes the lowest possible accuracy to satisfy fairness constraints. An allocation $\boldsymbol{a} = (a_1, \ldots, a_N) \in \mathcal{A} \equiv \{a_k\}^N$ assigns exactly one model with a performance level $a_i \in [\ell, u]$ to each client $i$. The gain (utility) of client $i$ under allocation $\boldsymbol{a}$ is defined as:

$$u_i(\boldsymbol{a}) = a_i - c_i. \tag{2}$$

An ideal allocation algorithm must satisfy the following three objectives:

1. **Individual Rationality (IR):** The gain of every client must be nonnegative, satisfying individual rationality (Definition 1) and ensuring the participation of all rational agents.

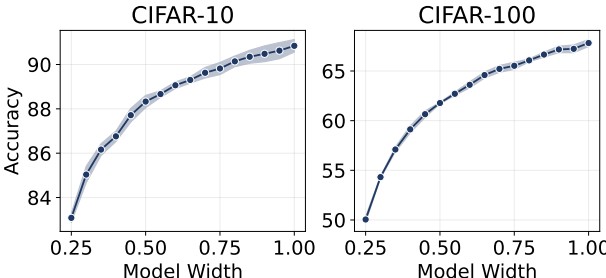

Figure 1: Performance vs. network width ($p$) using CIFAR-10 and CIFAR-100 datasets on ResNet-18 model.

2. **Low variability in utility:** The variability of utilities $\{u_i(\boldsymbol{a})\}$ should be kept low so that no client's benefit is disproportionately high or low compared to others.

3. **High average gain:** The average gain across clients, $\frac{1}{N} \sum_{i=1}^{N} u_i(\boldsymbol{a})$, should be as large as possible, balancing overall performance with fairness.

**Definition 1** (Individual Rationality (IR)). An allocation satisfies IR if $u_i(\boldsymbol{a}) \geq 0$ for all $i$. This ensures that no client is worse off than their standalone performance.

The above objectives can be achieved by maximizing the average gain $\mathbb{E}[u(\boldsymbol{a})]$, while simultaneously minimizing the variance $\mathrm{Var}[u(\boldsymbol{a})]$ and respecting the IR constraint. Formally, we define the optimization problem as minimizing the cost function:

$$f(\boldsymbol{a}) = -\frac{\mathbb{E}[u(\boldsymbol{a})]}{\mathrm{Var}[u(\boldsymbol{a})] + \epsilon}, \tag{3}$$

where $f : \mathcal{A} \to \mathbb{R}$, and $\epsilon > 0$ is a small constant introduced to prevent division by zero. To solve this constrained optimization problem, we employ a simulated annealing algorithm (Granville et al., 1994; Bouttier & Gavra, 2019) and adapt it to our specific setting.

### 4.3. Extension to Training-time Rewards

Thus far, our focus has been on post-training model rewards. However, our approach can be seamlessly extended to incorporate training-time rewards as well. The primary modification involves sending a slimmed-down network (a subnetwork) to each client during training, rather than the full model, and dynamically updating their contributions based on the quality of their shared updates.

A well-established method for the evaluation of contributions is CGSV (Xu et al., 2021), and our approach can leverage CGSV to evaluate the contribution of each participant. Additionally, other contribution assessment methods (CA), such as FedSV (Wang et al., 2020), GTG-Shapley (Liu et al., 2022b), ComFedSV (Fan et al., 2022), FedFAIM (Shi

et al., 2022), ShapFed (Tastan et al., 2024a), among others, can also be integrated within this framework.

Since some clients, typically low-performing clients, train only on subnetworks and do not share full gradients, they instead transmit their trained subnetwork updates. During the contribution assessment phase, we evaluate the gradients corresponding to the minimum-width subnetwork selected by the algorithm (e.g. $0.25\times$ as shown in Figure 1), ensuring consistency since all clients train on this common subnetwork.

Furthermore, any incentive mechanism can be employed to dynamically adjust client contributions. In our approach, since the maximum width assigned to each participant is determined on the basis of estimated contributions, we directly map the normalized contributions to network widths. This ensures that the highest contributor(s) receive the full model. The update rule of contributions is given by:

$$c_i^t = \gamma c_i^{t-1} + (1-\gamma)\tilde{c}_i, \qquad (4)$$

where $\gamma$ is a momentum parameter and $\tilde{c}_i$ is obtained using the CA method. The reward mechanism is then defined as:

$$[\mathcal{M}_\nu(\boldsymbol{c})]_i = \nu(c_i/\max_k c_k) \qquad (5)$$

where $\nu$ is a utility function that directly maps the contributions to network widths $(p_i)$.

We present experimental results for this approach in Figures 3 and 4. Further implementation details and a pseudo-algorithm outlining this extension are provided in Appendix C.

## 5. Theoretical Analysis

We now present the convergence analysis of the main algorithm described in Section 4.1, along with the convergence and fairness analysis of the allocation algorithm in Section 4.2.

### 5.1. Convergence Analysis

Formally, we establish the following assumptions for the convergence analysis, which pertain to the properties of local objectives – namely, $L$-smoothness (Assumption 1), convexity (Assumption 2), and bounded variance (Assumption 3, Appendix A.5.1). Additionally, we account for the data similarity across participants through bounded dissimilarity (Assumption 4, Appendix A.5.1).

**Assumption 1** ($L$-smoothness). The local objective $F_i(\boldsymbol{x}), \forall i \in [N]$ is $L$-smooth, then for all $\boldsymbol{x}, \boldsymbol{y} \in \mathbb{R}^d$,

$$F_i(\boldsymbol{x}) \leq F_i(\boldsymbol{y}) + \langle \nabla F_i(\boldsymbol{x}), \boldsymbol{y} - \boldsymbol{x} \rangle + \frac{L}{2}\|\boldsymbol{x} - \boldsymbol{y}\|^2,$$
$$\forall i \in [N].$$

**Assumption 2** (Convexity). The local objective $F_i(\boldsymbol{x}), \forall i \in [N]$ is both convex and $L$-smooth, then for all $\boldsymbol{x}, \boldsymbol{y}, \boldsymbol{z} \in \mathbb{R}^d$,

$$\frac{1}{L}\|\nabla F_i(\boldsymbol{x}) - \nabla F_i(\boldsymbol{y})\|^2 \leq \langle \nabla F_i(\boldsymbol{x}) - F_i(\boldsymbol{y}), \boldsymbol{x} - \boldsymbol{y} \rangle,$$

$$F_i(\boldsymbol{x}) \leq F_i(\boldsymbol{y}) + \langle \nabla F_i(\boldsymbol{z}), \boldsymbol{x} - \boldsymbol{y} \rangle + \frac{L}{2}\|\boldsymbol{x} - \boldsymbol{z}\|^2,$$
$$\forall i \in [N].$$

Next, we present the lemmas that establish the L-smoothness and convexity properties of the slimmed models.

**Lemma 1.** *Let $F : \mathbb{R}^d \to \mathbb{R}$ be an L-smooth function over $\mathbb{R}^d$. Consider a selection of $d_1 \leq d$ coordinates from $\boldsymbol{x}$, which we denote by $\widetilde{\boldsymbol{x}} \in \mathbb{R}^{d_1}$. Define*

$$F_{\boldsymbol{r}}(\widetilde{\boldsymbol{x}}) = F(\widetilde{\boldsymbol{x}}, \boldsymbol{r}), \qquad (6)$$

*where $\boldsymbol{r} \in \mathbb{R}^{d-d_1}$ contains the fixed (zeros) coordinates of $\boldsymbol{x}$. Then $F_{\boldsymbol{r}}$ is L-smooth in the subspace $\mathbb{R}^{d_1}$ with a constant $\widetilde{L} \leq L$.*

*Proof of Lemma 1.* See Appendix A.1 □

**Lemma 2.** *Let $F : \mathbb{R}^d \to \mathbb{R}$ be a convex function, as per Assumption 2. Consider a subset of coordinates $\widetilde{\boldsymbol{x}} \in \mathbb{R}^{d_1}$ (with $d_1 \leq d$) of the vector $\boldsymbol{x} \in \mathbb{R}^d$, and let $\boldsymbol{r} \in \mathbb{R}^{d-d_1}$ denote the remaining fixed coordinates. Define*

$$F_{\boldsymbol{r}}(\widetilde{\boldsymbol{x}}) = F(\widetilde{\boldsymbol{x}}, \boldsymbol{r}).$$

*Then $F_{\boldsymbol{r}}$ is convex in $\mathbb{R}^{d_1}$.*

*Proof of Lemma 2.* See Appendix A.2 □

With the established $L$-smoothness and convexity parameters of the slimmed models, we can now directly derive the convergence guarantee for Local SGD. For completeness, we present Theorem 1, which provides a performance guarantee under the given assumptions (Stich, 2018; Wang et al., 2021; Khaled et al., 2020; Woodworth et al., 2020).

**Theorem 1** (Performance guarantee). *Under the assumptions 1, 2, 3, 4, if the participant learning rate satisfies $\eta \leq \frac{1}{4L}$, then the performance is bounded by*

$$\mathbb{E}\left[\frac{1}{\tau T}\sum_{t=0}^{T-1}\sum_{k=1}^{\tau} F(\overline{\boldsymbol{x}}^{t,k}) - F(\boldsymbol{x}^\star)\right]$$

$$\leq \frac{D^2}{2\eta\tau T} + \frac{\eta\sigma^2}{N} + 4\tau\eta^2 L\sigma^2 + 18\tau^2\eta^2 L\zeta^2,$$

*where $D := \|\boldsymbol{x}^{0,0} - \boldsymbol{x}^\star\|$.*

*Proof of Theorem 1.* Taking into account Lemmas 1, 2, 5, and 6, combining them, and telescoping $t$ over $T$ communication rounds gives us the main theorem. □

## 5.2. Fairness Analysis

We now analyze the fairness properties of the proposed allocation algorithm, highlighting key theoretical results and their practical implications.

**Lemma 3.** *Let $\boldsymbol{a}^\star = (a_1, a_2, \ldots, a_N)$ be an optimal allocation that minimizes $f(\boldsymbol{a})$. Then, it holds that $\max_{i \in [N]} a_i = u$. In other words, the client with the highest contribution always receives the model with the maximum accuracy $u$.*

*Proof of Lemma 3.* See Appendix A.3. □

Lemma 3 ensures that the highest contributor is always rewarded with the most accurate model available, satisfying intuitive fairness criteria. Furthermore, under the assumption of continuous availability of model accuracies within the interval $[\ell, u]$, the optimal allocation aligns perfectly with client contributions, as shown by the following lemma:

**Lemma 4.** *Let $\boldsymbol{a}^\star$ be the optimal allocation minimizing $f(\boldsymbol{a})$, and $\boldsymbol{c}$ the vector of client contributions. Under the assumption of continuous model performance availability, the Pearson correlation coefficient between allocated accuracies $\boldsymbol{a}^\star$ and client contributions $\boldsymbol{c}$ is exactly one:*

$$\rho(\boldsymbol{a}^\star, \boldsymbol{c}) = 1.$$

*Proof of Lemma 4.* See Appendix A.4. □

Here, $\rho(\boldsymbol{a}^\star, \boldsymbol{c})$ denotes the Pearson correlation between vectors $\boldsymbol{a}^\star$ and $\boldsymbol{c}$, measuring how well the allocated rewards match the client contributions. A correlation of exactly one indicates perfect alignment, thus confirming that the proposed method achieves the theoretical ideal collaborative fairness (Lyu et al., 2020; Tastan et al., 2025) under continuous model performance conditions.

In practice, however, model widths and their associated performance values must be discretized due to inherent computational constraints, introducing slight deviations from perfect correlation. Despite this practical limitation, we consistently observe near-perfect Pearson correlation coefficients in empirical evaluations, as shown in Table 2.

## 5.3. Convergence of the Allocation Algorithm

As described in Section 4.2, we define a cost function

$$f : \mathcal{A} \to \mathbb{R}, \qquad (7)$$

which we aim to minimize over the space of allocations $\mathcal{A}$. Under suitable conditions, we can guarantee convergence of the allocation algorithm to a globally optimal solution.

We now present Theorem 2, which guarantees that, under assumptions 5-9 (provided in Appendix A.6.1), the allocation algorithm converges asymptotically to the global minimizer of the cost function $f$ with probability one.

**Theorem 2** (Asymptotic convergence). *Under Assumptions 5, 6, 7, 8, 9, consider the time-inhomogeneous Markov chain $\{A_k\}$ on $\mathcal{A}$ with transition probabilities defined by*

$$P_k(\boldsymbol{a} \to \boldsymbol{a}') = \begin{cases} \exp\left(-\dfrac{f(\boldsymbol{a}') - f(\boldsymbol{a})}{T_k}\right), & f(\boldsymbol{a}') > f(\boldsymbol{a}), \\ 1, & f(\boldsymbol{a}') \le f(\boldsymbol{a}), \end{cases}$$

*where $T_k$ is the temperature parameter at iteration $k$.*

*Then for any **global minimizer** $\boldsymbol{a}^\star \in \mathcal{A}$ satisfying $f(\boldsymbol{a}^\star) = \min_{\boldsymbol{a} \in \mathcal{A}} f(\boldsymbol{a})$, the following holds:*

$$\lim_{k \to \infty} P(A_k = \boldsymbol{a}^\star) = 1.$$

*In other words, the Markov chain $\{A_k\}$ converges (with probability 1) to the set of global minimizers of $f$.*

*Proof of Theorem 2.* See Appendix A.6. □

# 6. Experiments and Results

## 6.1. Experimental details

**Datasets and partition settings.** We use the following datasets to carry out our experiments (following (Li et al., 2020)): MNIST (LeCun, 1998), Fashion-MNIST (FMNIST) (Xiao et al., 2017), SVHN (Netzer et al., 2011), CIFAR-10 & CIFAR-100 (Krizhevsky et al., 2009), Stanford Sentiment Treebank (SST) (Socher et al., 2013), and the federated handwriting dataset FEMNIST (Caldas et al., 2019). FEMNIST is already partitioned by writer identities, yielding a naturally heterogeneous distribution that we keep unchanged. The other datasets are partitioned using the following strategies: (i) **homogeneous**, where each participant gets an equal number of data points per class; (ii) **heterogeneous**, where each client gets a varying number of data points per class based on a Dirichlet($\alpha$) distribution (concentration parameter $\alpha$ reflects the degree of non-i.i.d. characteristics within the dataset); (iii) **quantity skew** allocates $\kappa$ proportion of total data points to each of the $m$ selected participants and the remaining $N - m$ participants split the remaining data equally; (iv) **label skew**, denoted by $\#C = m$, creates a label imbalance by sampling $m$ classes for each client and then randomly distributing samples from class $m$ among selected participants.

**Baseline approaches.** We compare our approach to a FedAvg algorithm (McMahan et al., 2017) followed by an additional epoch of local training to obtain a differentiated model for each participant, CGSV (Xu et al., 2021) and IAFL (Wu et al., 2024). We also evaluate standalone accuracy (SA), where each client trains its ML model on its local dataset without any collaboration with others. For a detailed

Table 1: Predictive performance (%, higher is better) of AEQUA and baselines under different dataset partitioning regimes.

| PARTITION | DATASET | FEDAVG | CGSV | IAFL | AEQUA |
|---|---|---|---|---|---|
| HOMOGENEOUS | MNIST | **98.67** $\pm$ 0.07 | 90.62 $\pm$ 2.32 | 98.40 $\pm$ 0.13 | 98.60 $\pm$ 0.11 |
| | FMNIST | 89.45 $\pm$ 0.33 | 77.01 $\pm$ 2.51 | 88.54 $\pm$ 0.25 | **89.63** $\pm$ 0.19 |
| | SVHN | **90.54** $\pm$ 0.18 | 77.61 $\pm$ 3.16 | 89.64 $\pm$ 0.13 | 90.18 $\pm$ 0.15 |
| | CIFAR-10 | 89.99 $\pm$ 0.23 | 61.29 $\pm$ 2.92 | 88.42 $\pm$ 0.07 | **90.84** $\pm$ 0.26 |
| | CIFAR-100 | 65.92 $\pm$ 0.22 | 35.36 $\pm$ 0.77 | 63.23 $\pm$ 0.30 | **67.83** $\pm$ 0.32 |
| | SST | 34.44 $\pm$ 1.33 | 30.12 $\pm$ 1.03 | 34.02 $\pm$ 0.51 | **34.44** $\pm$ 1.19 |
| HETEROGENEOUS: DIRICHLET ($\alpha = 0.1$) | MNIST | **97.38** $\pm$ 0.62 | 94.06 $\pm$ 1.79 | 88.45 $\pm$ 7.87 | 97.30 $\pm$ 0.58 |
| | FMNIST | 83.32 $\pm$ 1.78 | 71.51 $\pm$ 8.20 | 66.46 $\pm$ 4.72 | **84.60** $\pm$ 1.32 |
| | SVHN | **86.38** $\pm$ 0.87 | 72.48 $\pm$ 4.81 | 68.71 $\pm$ 7.85 | 86.33 $\pm$ 1.00 |
| | CIFAR-10 | 74.73 $\pm$ 3.65 | 48.77 $\pm$ 5.02 | 46.36 $\pm$ 9.31 | **75.97** $\pm$ 3.36 |
| | CIFAR-100 | 61.16 $\pm$ 0.25 | 34.16 $\pm$ 1.63 | 43.38 $\pm$ 4.53 | **63.42** $\pm$ 0.54 |
| | SST | 32.17 $\pm$ 1.60 | 21.54 $\pm$ 1.89 | 27.28 $\pm$ 3.06 | **33.54** $\pm$ 1.48 |
| QUANTITY SKEW: IMBALANCED $(0.15, 6)$ | MNIST | **98.69** $\pm$ 0.10 | 93.22 $\pm$ 0.99 | 98.40 $\pm$ 0.12 | 98.62 $\pm$ 0.09 |
| | FMNIST | 89.53 $\pm$ 0.24 | 78.73 $\pm$ 2.14 | 88.53 $\pm$ 0.26 | **89.72** $\pm$ 0.17 |
| | SVHN | **90.59** $\pm$ 0.17 | 77.54 $\pm$ 2.34 | 89.51 $\pm$ 0.13 | 90.26 $\pm$ 0.16 |
| | CIFAR-10 | 90.00 $\pm$ 0.13 | 66.89 $\pm$ 1.85 | 89.51 $\pm$ 0.15 | **90.71** $\pm$ 0.14 |
| | CIFAR-100 | 65.88 $\pm$ 0.38 | 39.62 $\pm$ 0.74 | 64.60 $\pm$ 0.25 | **68.24** $\pm$ 0.11 |
| | SST | 34.26 $\pm$ 0.98 | 29.53 $\pm$ 1.02 | 33.70 $\pm$ 0.70 | **34.64** $\pm$ 1.01 |
| LABEL SKEW: #OC={3, 30} | MNIST | 94.37 $\pm$ 3.43 | 79.19 $\pm$ 7.94 | 73.10 $\pm$ 15.00 | **95.37** $\pm$ 1.15 |
| | FMNIST | 79.73 $\pm$ 3.80 | 61.54 $\pm$ 8.03 | 60.10 $\pm$ 8.03 | **80.51** $\pm$ 3.27 |
| | SVHN | 79.73 $\pm$ 5.89 | 64.07 $\pm$ 7.65 | 55.83 $\pm$ 11.89 | **80.69** $\pm$ 6.05 |
| | CIFAR-10 | 71.88 $\pm$ 3.28 | 48.02 $\pm$ 3.88 | 44.12 $\pm$ 21.15 | **72.40** $\pm$ 3.17 |
| | CIFAR-100 | 60.95 $\pm$ 1.18 | 35.09 $\pm$ 0.42 | 55.26 $\pm$ 3.85 | **62.84** $\pm$ 1.18 |
| | SST | **33.96** $\pm$ 0.35 | 24.88 $\pm$ 2.04 | 30.33 $\pm$ 1.68 | 33.01 $\pm$ 0.90 |
| NUMBER OF TIMES THAT PERFORMS THE BEST | | 7/24 | 0/24 | 0/24 | **17/24** |

explanation of the baseline approaches, evaluation metrics, and implementation details, we refer to Appendix B.

## 6.2. Predictive performance

We begin by benchmarking existing algorithms (FedAvg, CGSV, and IAFL) across the six datasets introduced in Section 6.1, using the partitioning strategies detailed in Table 1. Due to space constraints, we present a summarized version of the results, while full details can be found in Table 5, Appendix D.2. We report the accuracy (balanced accuracy) of the global model as an evaluation metric.

Aequa attains the highest mean accuracy in 17 of the 24 settings, while FedAvg leads in the remaining 7 settings. In contrast, the fairness-oriented baselines CGSV and IAFL lag markedly, especially under the more challenging heterogeneous partitioning settings (Dirichlet ($\alpha = 0.1$) and label skew). Their lower scores reflect an inherent performance-fairness trade-off. By adaptively allocating capacity, Aequa mitigates this trade-off and matches or exceeds FedAvg across all splits, corroborating the established performance guarantees. We also provide a per-participant performance analysis on the CIFAR-100 dataset in Appendix D.5.

## 6.3. Incentivization performance

**Correlation to contribution.** Following Wu et al. (2024), we quantify how well each algorithm aligns incentives with

actual usefulness by computing the Pearson correlation coefficient ($\rho$) between the client model accuracies achieved after $T$ communication rounds and their standalone accuracies. This metric also serves as a measure of fairness, as highlighted in (Xu et al., 2021; Lyu et al., 2020). Following IAFL (Wu et al., 2024), we use standalone accuracies as a surrogate for client contributions. Thus, by analyzing the correlation values, we can directly compare the incentivization effectiveness of different algorithms. A high positive $\rho$ indicates that clients who contribute stronger local models are rewarded with higher final accuracies, while values near zero or negative signal poor incentive alignment.

Table 2 summarizes the results. FedAvg-FT fails to establish a strong correlation, often yielding negative values, particularly under quantity-skewed splits, while CGSV and IAFL struggle whenever standalone accuracies show little variation (e.g., the homogeneous setting). IAFL improves under quantity skew, yet Aequa delivers both tighter and uniformly higher correlations, ranking first in 24/24 cases. For a comprehensive analysis, we refer the reader to Figure 6, Appendix D.3, where we present results in 54 different scenarios. In particular, our method outperforms all baselines in all cases, achieving a perfect score of 54/54.

**Collaboration gain spread.** We evaluate the collaboration gain spread (CGS) achieved by Aequa in comparison to other baseline methods. As reported in Table 7 in

Table 2: Incentivization performance comparison of our method and baseline approaches across different dataset partitions, evaluated using the Pearson correlation coefficient between the accuracies of the final model and the accuracies of the standalone model. The results are averaged over five independent evaluations. For complete results, refer to Appendix D.3.

| PARTITION | DATASET | FEDAVG-FT | CGSV | IAFL | AEQUA |
|---|---|---|---|---|---|
| | MNIST | $0.07 \pm 0.24$ | $-0.30 \pm 0.16$ | $0.16 \pm 0.20$ | $\mathbf{0.97 \pm 0.02}$ |
| | FMNIST | $0.21 \pm 0.07$ | $-0.09 \pm 0.35$ | $0.27 \pm 0.24$ | $\mathbf{0.98 \pm 0.02}$ |
| HOMOGENEOUS | SVHN | $-0.12 \pm 0.36$ | $0.02 \pm 0.17$ | $0.07 \pm 0.28$ | $\mathbf{0.98 \pm 0.02}$ |
| | CIFAR-10 | $0.04 \pm 0.27$ | $0.05 \pm 0.42$ | $-0.01 \pm 0.15$ | $\mathbf{0.99 \pm 0.01}$ |
| | CIFAR-100 | $-0.07 \pm 0.37$ | $-0.19 \pm 0.36$ | $0.02 \pm 0.31$ | $\mathbf{0.96 \pm 0.01}$ |
| | SST | $0.06 \pm 0.25$ | $-0.07 \pm 0.32$ | $-0.03 \pm 0.26$ | $\mathbf{0.98 \pm 0.01}$ |
| | MNIST | $0.39 \pm 0.37$ | $0.56 \pm 0.18$ | $0.61 \pm 0.23$ | $\mathbf{0.85 \pm 0.03}$ |
| | FMNIST | $0.21 \pm 0.41$ | $0.56 \pm 0.28$ | $0.61 \pm 0.25$ | $\mathbf{0.89 \pm 0.06}$ |
| HETEROGENEOUS: | SVHN | $0.66 \pm 0.24$ | $0.21 \pm 0.46$ | $0.80 \pm 0.18$ | $\mathbf{0.92 \pm 0.02}$ |
| DIRICHLET ($\alpha = 0.1$) | CIFAR-10 | $-0.18 \pm 0.35$ | $0.67 \pm 0.14$ | $0.84 \pm 0.15$ | $\mathbf{0.94 \pm 0.02}$ |
| | CIFAR-100 | $-0.17 \pm 0.46$ | $0.30 \pm 0.52$ | $0.89 \pm 0.07$ | $\mathbf{0.99 \pm 0.01}$ |
| | SST | $-0.20 \pm 0.35$ | $0.14 \pm 0.50$ | $0.91 \pm 0.09$ | $\mathbf{0.98 \pm 0.01}$ |
| | MNIST | $-0.63 \pm 0.18$ | $0.34 \pm 0.80$ | $0.95 \pm 0.04$ | $\mathbf{0.98 \pm 0.01}$ |
| | FMNIST | $-0.45 \pm 0.28$ | $0.49 \pm 0.71$ | $0.93 \pm 0.02$ | $\mathbf{0.98 \pm 0.01}$ |
| QUANTITY SKEW: | SVHN | $-0.76 \pm 0.12$ | $0.42 \pm 0.75$ | $0.99 \pm 0.01$ | $\mathbf{1.00 \pm 0.00}$ |
| IMBALANCED $(0.15, 6)$ | CIFAR-10 | $-0.37 \pm 0.16$ | $0.98 \pm 0.02$ | $0.99 \pm 0.00$ | $\mathbf{1.00 \pm 0.00}$ |
| | CIFAR-100 | $0.06 \pm 0.40$ | $0.97 \pm 0.03$ | $\mathbf{1.00 \pm 0.00}$ | $\mathbf{1.00 \pm 0.00}$ |
| | SST | $-0.07 \pm 0.48$ | $-0.23 \pm 0.54$ | $0.90 \pm 0.02$ | $\mathbf{0.94 \pm 0.04}$ |
| | MNIST | $0.03 \pm 0.41$ | $-0.27 \pm 0.27$ | $0.23 \pm 0.24$ | $\mathbf{0.81 \pm 0.13}$ |
| | FMNIST | $-0.44 \pm 0.27$ | $0.11 \pm 0.45$ | $0.08 \pm 0.26$ | $\mathbf{0.99 \pm 0.01}$ |
| LABEL SKEW: #OC=$\{3, 30\}$ | SVHN | $0.43 \pm 0.25$ | $-0.43 \pm 0.42$ | $0.01 \pm 0.25$ | $\mathbf{0.98 \pm 0.00}$ |
| | CIFAR-10 | $0.19 \pm 0.32$ | $0.12 \pm 0.32$ | $0.22 \pm 0.38$ | $\mathbf{0.97 \pm 0.02}$ |
| | CIFAR-100 | $-0.38 \pm 0.22$ | $0.00 \pm 0.24$ | $0.31 \pm 0.22$ | $\mathbf{0.98 \pm 0.02}$ |
| | SST | $0.45 \pm 0.38$ | $-0.19 \pm 0.48$ | $0.48 \pm 0.37$ | $\mathbf{0.97 \pm 0.03}$ |
| NUMBER OF TIMES THAT PERFORMS THE BEST | | 0/24 | 0/24 | 1/24 | **24/24** |

Table 3: Comparison to baseline methods when the contribution measure corresponds to the participation rate.

| METHOD | MNIST | | CIFAR-10 | |
|---|---|---|---|---|
| | $\rho$ | ACC. | $\rho$ | ACC. |
| FEDAVG-FT | $0.12 \pm 0.1$ | $97.65 \pm 0.3$ | $0.11 \pm 0.1$ | $83.73 \pm 0.3$ |
| CGSV | $0.50 \pm 0.2$ | $96.18 \pm 0.7$ | $0.55 \pm 0.1$ | $55.97 \pm 0.3$ |
| IAFL | $0.79 \pm 0.0$ | $98.04 \pm 0.2$ | $0.69 \pm 0.2$ | $82.58 \pm 0.1$ |
| AEQUA | $\mathbf{0.98 \pm 0.0}$ | $\mathbf{98.19 \pm 0.1}$ | $\mathbf{0.99 \pm 0.0}$ | $\mathbf{84.59 \pm 0.2}$ |

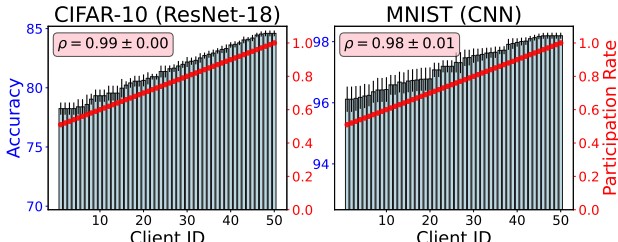

Figure 2: Visualization of the correlation when contribution measure corresponds to the participation rate on CIFAR-10 and MNIST datasets with $N = 50$ participants.

Appendix D.4, Aequa consistently outperforms other approaches in 42 out of 54 scenarios, demonstrating consistently superior collaboration dynamics.

One scenario in which Aequa underperforms relative to IAFL is in the quantity skew setting. This is due to the choice of $p_{\min}$, which determines the minimum accuracy level $\ell$ assigned to low-contributing clients. To address this, we conducted an additional set of experiments, adjusting $p_{\min}$ to 0.1, presented in Table 8. Under this configuration, Aequa outperforms other methods in 11/12 cases, bringing the overall total to 48/54 cases.

### 6.4. Alternative contribution measures

Aequa is designed to flexibly incorporate a wide range of contribution measures. To demonstrate its effectiveness

under partial participation, we use the client participation rate as a contribution measure. Following the experimental setup studied in (Wu et al., 2024), we define the client participation rate as $r_i = 0.5 \times (1 + i/N)$, introducing a mild but controlled variation across clients, as illustrated in Figure 2.

We conducted experiments on MNIST and CIFAR-10 using distinct architectures to assess Aequa's robustness. As shown in Figure 2, our method yields a very high Pearson correlation coefficient, demonstrating a strong alignment between participation rates and the resulting model performance. Table 3 presents a comparative analysis with other baseline algorithms, which confirms that Aequa consistently

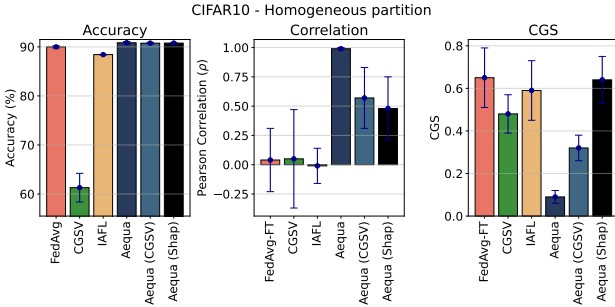

Figure 3: Performance comparison of our proposed methods on CIFAR-10 under a homogeneous partitioning strategy.

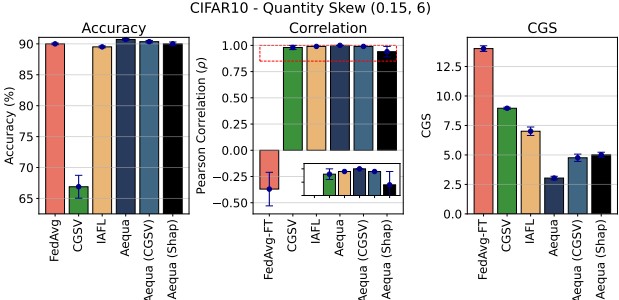

Figure 4: Performance comparison of our proposed methods on CIFAR-10 under a quantity skew.

outperforms competitors in both correlation and final performance. In particular, IAFL is the second-best method in this setting, yet it trails Aequa by a clear margin.

### 6.5. Training-time model rewards

We evaluate our method in combination with existing contribution assessment algorithms for training-time model rewards, comparing against baseline methods. The results, presented in Figures 3 and 4, illustrate the performance on the CIFAR-10 dataset under homogeneous and quantity skew partitioning strategies. The plot presents all evaluation metrics, and the results clearly demonstrate that Aequa consistently outperforms other approaches across all cases. Among the variants, Aequa with CGSV ranks as the second-best in both settings, while Aequa with ShapFed shows comparable performance, further validating the robustness of our approach.

### 6.6. Scalability to large-scale FL

To probe the limits of Aequa under realistic federated conditions, we performed an experiment on the FEMNIST dataset (Caldas et al., 2019) – a challenging benchmark comprising $3\,597$ naturally non-IID users and more than $800$ thousand handwritten character images.

Table 4 shows that Aequa not only improves test accuracy

Table 4: Predictive and incentivization performance of Aequa in comparison to FedAvg-FT on FEMNIST dataset.

| ALGORITHM | ACCURACY | PEARSON ($\rho$) | MCG±CGS |
|---|---|---|---|
| FEDAVG-FT | 71.48 | 0.2904 | $47.02 \pm 5.07$ |
| AEQUA | **73**.10 | **0.9888** | $52.19 \pm$ **2.57** |

by $+1.6$ pp over the FedAvg-FT baseline but also yields markedly better fairness scores (Pearson correlation coefficient of $0.9888(\uparrow)$ and CGS of $2.57(\downarrow)$), confirming its ability to handle both data heterogeneity and the scale of thousands of clients.

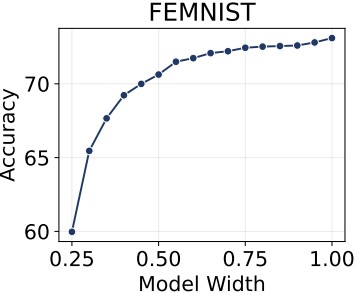

Figure 5: Performance vs. network width ($p$) using FEMNIST dataset using a custom CNN architecture composed of two convolutional layers followed by a fully connected layer.

Figure 5 further plots accuracy versus model width on FEMNIST, illustrating that Aequa's advantage is consistent across a wide range of capacities and that the resulting accuracy difference is sufficiently large to discourage free-riding behavior. These findings suggest that the proposed mechanism scales to large, real-world FL deployments.

**Computational overhead.** On the same FEMNIST setup, we recorded wall-clock training time per communication round on a single GPU. FedAvg finishes a round in $42\pm2.9$ s, whereas Aequa takes $55 \pm 3.1$ s – only a $1.309\times (\approx 30.9\%)$ increase that is modest, not significant and well justified by Aequa's superior performance and fairness. Communication cost is unchanged because both methods transmit identically sized model parameters.

## 7. Conclusion

Using the concept of slimmable networks, we have presented Aequa, a framework for achieving a provably fair model rewards in collaborative learning. While we initially present TEEs as a means to ensure the confidentiality of local updates, TEEs are not a strict requirement. Aequa also operates securely in the absence of trusted hardware by incorporating contribution assessment methods.

## Acknowledgements

This material is partly based on work supported by the Office of Naval Research N00014-24-1-2168.

## Impact Statement

This paper presents work whose goal is to advance the field of Machine Learning. There are many potential societal consequences of our work, none of which we feel must be specifically highlighted here.

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

# A. Mathematical proofs

## A.1. Proof of Lemma 1

*Proof of Lemma 1.* Since $F$ is $L$-smooth over $\mathbb{R}^d$, for any $\boldsymbol{x}, \boldsymbol{x}' \in \mathbb{R}^d$ we have

$$\|\nabla F(\boldsymbol{x}) - \nabla F(\boldsymbol{x}')\| \leq L\|\boldsymbol{x} - \boldsymbol{x}'\|. \tag{8}$$

Let $\widetilde{\boldsymbol{x}}, \widetilde{\boldsymbol{x}}' \in \mathbb{R}^{d_1}$ be two points in the reduced space of selected coordinates. We embed each into $\mathbb{R}^d$ by defining

$$\boldsymbol{x} = (\widetilde{\boldsymbol{x}}, \boldsymbol{r}), \quad \boldsymbol{x}' = (\widetilde{\boldsymbol{x}}', \boldsymbol{r}), \tag{9}$$

where $\boldsymbol{r}$ is held fixed in both cases (e.g. zeros in our case). Note that

$$\|\boldsymbol{x} - \boldsymbol{x}'\| = \|(\widetilde{\boldsymbol{x}}, \boldsymbol{r}) - (\widetilde{\boldsymbol{x}}', \boldsymbol{r})\| = \|\widetilde{\boldsymbol{x}} - \widetilde{\boldsymbol{x}}'\|. \tag{10}$$

Next, write the gradient of $F$ in coordinates:

$$\nabla F(\boldsymbol{x}) = (\nabla_{\widetilde{\boldsymbol{x}}} F(\boldsymbol{x}), \nabla_{\boldsymbol{r}} F(\boldsymbol{x})), \quad \nabla F(\boldsymbol{x}') = (\nabla_{\widetilde{\boldsymbol{x}}'} F(\boldsymbol{x}'), \nabla_{\boldsymbol{r}} F(\boldsymbol{x}')), \tag{11}$$

Because $\boldsymbol{x}$ and $\boldsymbol{x}'$ differ only in the $\widetilde{\boldsymbol{x}}$-coordinates (the $\boldsymbol{r}$-part is the same), it follows that

$$\|\nabla F_{\boldsymbol{r}}(\widetilde{\boldsymbol{x}}) - \nabla F_{\boldsymbol{r}}(\widetilde{\boldsymbol{x}}')\|^2 = \|\nabla_{\widetilde{\boldsymbol{x}}} F(\boldsymbol{x}) - \nabla_{\widetilde{\boldsymbol{x}}'} F(\boldsymbol{x}'))\|^2 \leq \|\nabla F(\boldsymbol{x}) - \nabla F(\boldsymbol{x}')\|^2. \tag{12}$$

By the $L$-smoothness of $F$ in $\mathbb{R}^d$ and putting all together, we have

$$\|\nabla F_{\boldsymbol{r}}(\widetilde{\boldsymbol{x}}) - \nabla F_{\boldsymbol{r}}(\widetilde{\boldsymbol{x}}')\| = \|\nabla_{\widetilde{\boldsymbol{x}}} F(\widetilde{\boldsymbol{x}}) - \nabla_{\widetilde{\boldsymbol{x}}'} F(\widetilde{\boldsymbol{x}}')\| \leq \|\nabla F(\boldsymbol{x}) - \nabla F(\boldsymbol{x}')\| \leq L\|\boldsymbol{x} - \boldsymbol{x}'\| = L\|\widetilde{\boldsymbol{x}} - \widetilde{\boldsymbol{x}}'\|. \tag{13}$$

Hence $F_{\boldsymbol{r}}$ (the restriction of $F$ to the subset $\widetilde{\boldsymbol{x}}$) is also $L$-smooth with respect to $\widetilde{\boldsymbol{x}}$. In particular, its smoothness constant $\widetilde{L}$ satisfies $\widetilde{L} \leq L$.

This concludes the proof.

$\square$

## A.2. Proof of Lemma 2

*Proof of Lemma 2.* Since $F$ is convex over $\mathbb{R}^d$, for any two points $\boldsymbol{x}, \boldsymbol{y} \in \mathbb{R}^d$ and any $\lambda \in [0, 1]$,

$$F(\lambda \boldsymbol{x} + (1 - \lambda)\boldsymbol{y}) \leq \lambda F(\boldsymbol{x}) + (1 - \lambda)F(\boldsymbol{y}). \tag{14}$$

Now fix $\boldsymbol{r} \in \mathbb{R}^{d-d_1}$.

Let $\widetilde{\boldsymbol{x}}, \widetilde{\boldsymbol{x}}' \in \mathbb{R}^{d_1}$ and $\lambda \in [0, 1]$. Then,

$$F_{\boldsymbol{r}}(\lambda \widetilde{\boldsymbol{x}} + (1 - \lambda)\widetilde{\boldsymbol{x}}') = F(\lambda(\widetilde{\boldsymbol{x}}, \boldsymbol{r}) + (1 - \lambda)(\widetilde{\boldsymbol{x}}', \boldsymbol{r})). \tag{15}$$

Since $\boldsymbol{r}$ is fixed, the interpolation in $\mathbb{R}^d$ satisfies

$$\lambda(\widetilde{\boldsymbol{x}}, \boldsymbol{r}) + (1 - \lambda)(\widetilde{\boldsymbol{x}}', \boldsymbol{r}) = (\lambda \widetilde{\boldsymbol{x}} + (1 - \lambda)\widetilde{\boldsymbol{x}}', \boldsymbol{r}). \tag{16}$$

By convexity of $F$ in $\mathbb{R}^d$,

$$F\left(\lambda(\widetilde{\boldsymbol{x}}, \boldsymbol{r}) + (1 - \lambda)(\widetilde{\boldsymbol{x}}', \boldsymbol{r})\right) \leq \lambda F(\widetilde{\boldsymbol{x}}, \boldsymbol{r}) + (1 - \lambda)F(\widetilde{\boldsymbol{x}}', \boldsymbol{r}). \tag{17}$$

Rewriting in terms of $F_{\boldsymbol{r}}$,

$$F_{\boldsymbol{r}}(\lambda \widetilde{\boldsymbol{x}} + (1 - \lambda)\widetilde{\boldsymbol{x}}') \leq \lambda F_{\boldsymbol{r}}(\widetilde{\boldsymbol{x}}) + (1 - \lambda)F_{\boldsymbol{r}}(\widetilde{\boldsymbol{x}}'). \tag{18}$$

Hence $F_{\boldsymbol{r}}$ is convex in $\mathbb{R}^{d_1}$. This completes the proof.

$\square$

### A.3. Proof of Lemma 3

*Proof.* Suppose, for contradiction, that in the optimal allocation $\boldsymbol{a}^\star$, we have $a_N^\star < u$. Let $\alpha = u - a_N^\star > 0$. Because the set of achievable accuracies is assumed to be *continuous* on $[\ell, u]$, we can attempt to shift all the assigned accuracies in $\boldsymbol{a}^\star$ upward by $\alpha$. Define the new allocation $\widetilde{\boldsymbol{a}} = (\widetilde{a}_1, \widetilde{a}_2, \ldots, \widetilde{a}_N)$ by $\widetilde{\boldsymbol{a}} = \boldsymbol{a}^\star + \alpha\mathbf{1} \in \mathcal{A}$.

Then, this gives us

$$
\begin{aligned}
f(\widetilde{\boldsymbol{a}}) \;=\; & f(\boldsymbol{a}^\star + \alpha\mathbf{1}) = -\frac{\mathbb{E}[u(\boldsymbol{a}^\star)] + \alpha}{\mathrm{Var}[u(\boldsymbol{a}^\star)] + \epsilon} \\
< \; & -\frac{\mathbb{E}[u(\boldsymbol{a}^\star)]}{\mathrm{Var}[u(\boldsymbol{a}^\star)] + \epsilon} = f(\boldsymbol{a}^\star),
\end{aligned}
$$

contradicting the optimality of $\boldsymbol{a}^\star$. Thus, we must have $\alpha = 0$ and therefore $a_N^\star = u$.

$\square$

### A.4. Proof of Lemma 4

*Proof.* Since model performances are continuous on the interval $[\ell, u]$ and that $a_N^\star = u$, we define $\alpha = u - c_N > 0$. By construction, for every client $i$, the allocation satisfies $a_i^\star = c_i + \alpha$. Consequently, the Pearson correlation coefficient is computed as:

$$
\rho(\boldsymbol{a}^\star, \boldsymbol{c}) = \rho(\boldsymbol{c} + \alpha\mathbf{1}, \boldsymbol{c}) = \rho(\boldsymbol{c}, \boldsymbol{c}) = 1. \tag{19}
$$

This proves the lemma.

$\square$

### A.5. Deferred Lemmas

#### A.5.1. ASSUMPTIONS

**Assumption 3** (Expected stochastic gradient variance)**.** The variance of an unbiased stochastic gradient in participant is $\sigma^2$-uniformly bounded in $L_2$ norm, $\forall i \in [N], \forall k \in [\tau], \forall t \in [T]$,

$$
\mathbb{E}\left[g_i(\boldsymbol{x}_i^{t,k}) \mid \boldsymbol{x}_i^{t,k}\right] = \nabla F_i(\boldsymbol{x}_i^{t,k}),
$$

$$
\mathbb{E}\left[\left\|g_i(\boldsymbol{x}_i^{t,k}) - \nabla F_i(\boldsymbol{x}_i^{t,k})\right\|^2 \Big| \boldsymbol{x}_i^{t,k}\right] \leq \sigma^2.
$$

**Assumption 4** (Gradient dissimilarity)**.** The difference of local gradient $\nabla F_i(\boldsymbol{x})$ and the global gradient $\nabla F(\boldsymbol{x})$ is $\zeta$-uniformly bounded in $L_2$ norm, $\forall i \in [N], \forall k \in [\tau], \forall t \in [T]$,

$$
\max_i \sup_{\boldsymbol{x}} \left\|\nabla F_i(\boldsymbol{x}_i^{t,k}) - \nabla F(\boldsymbol{x}_i^{t,k})\right\| \leq \zeta.
$$

#### A.5.2. LEMMAS

**Lemma 5.** *Assuming the participant learning rate satisfies $\eta \leq \frac{1}{4L}$, then*

$$
\begin{aligned}
\mathbb{E}\left[\frac{1}{\tau}\sum_{k=1}^{\tau} F(\overline{\boldsymbol{x}}^{t,k}) - F(\boldsymbol{x}^\star)\Big| \mathcal{F}^{t,0}\right] \leq & \frac{1}{2\eta\tau}\left(\left\|\overline{\boldsymbol{x}}^{t,0} - \boldsymbol{x}^\star\right\|^2 - \mathbb{E}\left[\left\|\overline{\boldsymbol{x}}^{t,\tau} - \boldsymbol{x}^\star\right\|^2 \Big| \mathcal{F}^{t,0}\right]\right) \\
& + \frac{\eta\sigma^2}{N} + \frac{L}{N\tau}\sum_{i=1}^{N}\sum_{k=0}^{\tau-1}\mathbb{E}\left[\left\|\boldsymbol{x}_i^{t,k} - \overline{\boldsymbol{x}}^{t,k}\right\|^2 \Big| \mathcal{F}^{t,0}\right],
\end{aligned}
$$

*where $\mathcal{F}^{t,0}$ is the $\sigma$-field representing all the historical information up to the start of the $t$-th round.*

*Proof of Lemma 5.*

$$\frac{1}{N} \sum_{i=1}^{N} \left\langle g_i(\boldsymbol{x}_i^{t,k}), \overline{\boldsymbol{x}}^{t,k+1} - \boldsymbol{x}^\star \right\rangle = \left\langle -\frac{1}{\eta} \left( \overline{\boldsymbol{x}}^{t,k+1} - \overline{\boldsymbol{x}}^{t,k} \right), \overline{\boldsymbol{x}}^{t,k+1} - \boldsymbol{x}^\star \right\rangle \tag{20}$$

$$= \frac{1}{2\eta} \left( \left\| \overline{\boldsymbol{x}}^{t,k} - \boldsymbol{x}^\star \right\|^2 - \left\| \overline{\boldsymbol{x}}^{t,k+1} - \overline{\boldsymbol{x}}^{t,k} \right\|^2 - \left\| \overline{\boldsymbol{x}}^{t,k+1} - \boldsymbol{x}^\star \right\|^2 \right). \tag{21}$$

where (20) uses the update rule $\overline{\boldsymbol{x}}^{t,k+1} = \overline{\boldsymbol{x}}^{t,k} - \eta \frac{1}{N} \sum_{i=1}^{N} g_i(\boldsymbol{x}_i^{t,k})$, (21) uses the parallelogram law, which is $\langle \boldsymbol{u}, \boldsymbol{v} \rangle = \frac{1}{2} \left( \|\boldsymbol{u}\|^2 + \|\boldsymbol{v}\|^2 - \|\boldsymbol{u} - \boldsymbol{v}\|^2 \right), \forall \boldsymbol{u}, \boldsymbol{v} \in \mathbb{R}^d$.

By assumptions 1 and 2, we have:

$$F_i(\overline{\boldsymbol{x}}^{t,k+1}) \leq F_i(\boldsymbol{x}_i^{t,k}) + \left\langle \nabla F_i(\boldsymbol{x}_i^{t,k}), \overline{\boldsymbol{x}}^{t,k+1} - \boldsymbol{x}_i^{t,k} \right\rangle + \frac{L}{2} \left\| \overline{\boldsymbol{x}}^{t,k+1} - \boldsymbol{x}_i^{t,k} \right\|^2 \tag{22}$$

$$\leq F_i(\boldsymbol{x}^\star) + \left\langle \nabla F_i(\boldsymbol{x}_i^{t,k}), \overline{\boldsymbol{x}}^{t,k+1} - \boldsymbol{x}^\star \right\rangle + \frac{L}{2} \left\| \overline{\boldsymbol{x}}^{t,k+1} - \boldsymbol{x}_i^{t,k} \right\|^2 \tag{23}$$

$$= F_i(\boldsymbol{x}^\star) + \left\langle \nabla F_i(\boldsymbol{x}_i^{t,k}), \overline{\boldsymbol{x}}^{t,k+1} - \boldsymbol{x}^\star \right\rangle + \frac{L}{2} \left\| (\overline{\boldsymbol{x}}^{t,k+1} - \overline{\boldsymbol{x}}^{t,k}) - (\boldsymbol{x}_i^{t,k} - \overline{\boldsymbol{x}}^{t,k}) \right\|^2 \tag{24}$$

$$\leq F_i(\boldsymbol{x}^\star) + \left\langle \nabla F_i(\boldsymbol{x}_i^{t,k}), \overline{\boldsymbol{x}}^{t,k+1} - \boldsymbol{x}^\star \right\rangle + L \left\| \overline{\boldsymbol{x}}^{t,k+1} - \overline{\boldsymbol{x}}^{t,k} \right\|^2 + L \left\| \boldsymbol{x}_i^{t,k} - \overline{\boldsymbol{x}}^{t,k} \right\|^2 \tag{25}$$

where (22) uses the $L$-smoothness property (see Assumption 1), (23) uses the three points descent lemma, which holds true when $F_i(\boldsymbol{x})$ is both convex and $L$-smooth (see Assumption 2), (24) includes the addition and subtraction of $\overline{\boldsymbol{x}}^{t,k}$ to the third term, and (25) applies this inequality $\|\boldsymbol{u} + \boldsymbol{v}\|^2 \leq 2(\|\boldsymbol{u}\|^2 + \|\boldsymbol{v}\|^2), \forall \boldsymbol{u}, \boldsymbol{v} \in \mathbb{R}^d$.

By combining (21) and (25), we get:

$$F(\overline{\boldsymbol{x}}^{t,k+1}) - F(\boldsymbol{x}^\star) = \frac{1}{N} \sum_{i=1}^{N} \left( F_i(\overline{\boldsymbol{x}}^{t,k+1}) - F(\boldsymbol{x}^\star) \right) \tag{26}$$

$$\leq \frac{1}{N} \sum_{i=1}^{N} \left\langle \nabla F_i(\boldsymbol{x}_i^{t,k}) - g_i(\boldsymbol{x}_i^{t,k}), \overline{\boldsymbol{x}}^{t,k+1} - \boldsymbol{x}^\star \right\rangle + L \left\| \overline{\boldsymbol{x}}^{t,k+1} - \overline{\boldsymbol{x}}^{t,k} \right\|^2$$

$$+ \frac{L}{N} \sum_{i=1}^{N} \left\| \boldsymbol{x}_i^{t,k} - \overline{\boldsymbol{x}}^{t,k} \right\|^2 + \frac{1}{N} \sum_{i=1}^{N} \left\langle g_i(\boldsymbol{x}_i^{t,k}), \overline{\boldsymbol{x}}^{t,k+1} - \boldsymbol{x}^\star \right\rangle \tag{27}$$

$$= \frac{1}{N} \sum_{i=1}^{N} \left\langle \nabla F_i(\boldsymbol{x}_i^{t,k}) - g_i(\boldsymbol{x}_i^{t,k}), \overline{\boldsymbol{x}}^{t,k+1} - \boldsymbol{x}^\star \right\rangle + L \left\| \overline{\boldsymbol{x}}^{t,k+1} - \overline{\boldsymbol{x}}^{t,k} \right\|^2$$

$$+ \frac{L}{N} \sum_{i=1}^{N} \left\| \boldsymbol{x}_i^{t,k} - \overline{\boldsymbol{x}}^{t,k} \right\|^2 + \frac{1}{2\eta} \left( \left\| \overline{\boldsymbol{x}}^{t,k} - \boldsymbol{x}^\star \right\|^2 - \left\| \overline{\boldsymbol{x}}^{t,k+1} - \overline{\boldsymbol{x}}^{t,k} \right\|^2 - \left\| \overline{\boldsymbol{x}}^{t,k+1} - \boldsymbol{x}^\star \right\|^2 \right) \tag{28}$$

where (27) adds and subtracts $\frac{1}{N} \sum_{i=1}^{N} \left\langle g_i(\boldsymbol{x}_i^{t,k}), \overline{\boldsymbol{x}}^{t,k+1} - \boldsymbol{x}^\star \right\rangle$, (28) replaces the last term with (21).

Since $\mathbb{E}\left[\nabla F_i(\boldsymbol{x}_i^{t,k}) - g_i(\boldsymbol{x}_i^{t,k})\Big|\mathcal{F}^{t,k}\right] = 0$ we have

$$\mathbb{E}\left[\frac{1}{N}\sum_{i=1}^{N}\left\langle\nabla F_i(\boldsymbol{x}_i^{t,k}) - g_i(\boldsymbol{x}_i^{t,k}), \overline{\boldsymbol{x}}^{t,k+1} - \boldsymbol{x}^\star\right\rangle\Bigg|\mathcal{F}^{t,k}\right] \tag{29}$$

$$= \mathbb{E}\left[\frac{1}{N}\sum_{i=1}^{N}\left\langle\nabla F_i(\boldsymbol{x}_i^{t,k}) - g_i(\boldsymbol{x}_i^{t,k}), \overline{\boldsymbol{x}}^{t,k+1} - \overline{\boldsymbol{x}}^{t,k}\right\rangle\Bigg|\mathcal{F}^{t,k}\right] \tag{30}$$

$$\leq \eta\mathbb{E}\left[\left\|\frac{1}{N}\sum_{i=1}^{N}\left(\nabla F_i(\boldsymbol{x}_i^{t,k}) - g_i(\boldsymbol{x}_i^{t,k})\right)\right\|^2\Bigg|\mathcal{F}^{t,k}\right] + \frac{1}{4\eta}\mathbb{E}\left[\left\|\overline{\boldsymbol{x}}^{t,k+1} - \overline{\boldsymbol{x}}^{t,k}\right\|^2\Big|\mathcal{F}^{t,k}\right] \tag{31}$$

$$\leq \frac{\eta\sigma^2}{N} + \frac{1}{4\eta}\mathbb{E}\left[\left\|\overline{\boldsymbol{x}}^{t,k+1} - \overline{\boldsymbol{x}}^{t,k}\right\|^2\Big|\mathcal{F}^{t,k}\right], \tag{32}$$

where (31) uses Young's inequality, which is $\langle\boldsymbol{u}, \boldsymbol{v}\rangle \leq \epsilon\|\boldsymbol{u}\|^2 + \frac{1}{4\epsilon}\|\boldsymbol{v}\|^2, \forall\epsilon > 0$ and $\forall\boldsymbol{u}, \boldsymbol{v} \in \mathbb{R}^d$, and (32) uses bounded covariance assumption (see Assumption 3) and independence across clients.

By plugging (32) back to the conditional expectation of (28) with $\eta \leq \frac{1}{4L}$, we get:

$$\mathbb{E}\left[F(\overline{\boldsymbol{x}}^{t,k+1}) - F(\boldsymbol{x}^\star)\right] + \frac{1}{2\eta}\left(\mathbb{E}\left[\left\|\overline{\boldsymbol{x}}^{t,k+1} - \boldsymbol{x}^\star\right\|^2\Big|\mathcal{F}^{t,k}\right] - \left\|\overline{\boldsymbol{x}}^{t,k} - \boldsymbol{x}^\star\right\|^2\right)$$

$$\leq \frac{\eta\sigma^2}{N} - \left(\frac{1}{4\eta} - L\right)\mathbb{E}\left[\left\|\overline{\boldsymbol{x}}^{t,k+1} - \overline{\boldsymbol{x}}^{t,k}\right\|^2\Big|\mathcal{F}^{t,k}\right] + \frac{L}{N}\sum_{i=1}^{N}\left\|\boldsymbol{x}_i^{t,k} - \overline{\boldsymbol{x}}^{t,k}\right\|^2 \tag{33}$$

$$\leq \frac{\eta\sigma^2}{N} + \frac{L}{N}\sum_{i=1}^{N}\left\|\boldsymbol{x}_i^{t,k} - \overline{\boldsymbol{x}}^{t,k}\right\|^2 \tag{34}$$

where (34) holds true since $\eta \leq \frac{1}{4L}$.

Telescoping $k$ from 0 to $\tau$ gives us:

$$\mathbb{E}\left[\frac{1}{\tau}\sum_{k=1}^{\tau}F(\overline{\boldsymbol{x}}^{t,k}) - F(\boldsymbol{x}^\star)\Bigg|\mathcal{F}^{t,0}\right] \leq \frac{1}{2\eta\tau}\left(\left\|\overline{\boldsymbol{x}}^{t,0} - \boldsymbol{x}^\star\right\|^2 - \mathbb{E}\left[\left\|\overline{\boldsymbol{x}}^{t,\tau} - \boldsymbol{x}^\star\right\|^2\Big|\mathcal{F}^{t,0}\right]\right)$$

$$+ \frac{\eta\sigma^2}{N} + \frac{L}{N\tau}\sum_{i=1}^{N}\sum_{k=0}^{\tau-1}\mathbb{E}\left[\left\|\boldsymbol{x}_i^{t,k} - \overline{\boldsymbol{x}}^{t,k}\right\|^2\Big|\mathcal{F}^{t,0}\right],$$

which completes the proof.

$\square$

**Lemma 6.** *Assuming the client learning rate satisfies $\eta \leq \frac{1}{4L}$, then*

$$\mathbb{E}\left[\left\|\boldsymbol{x}_i^{t,k} - \overline{\boldsymbol{x}}^{t,k}\right\|^2\Big|\mathcal{F}^{t,0}\right] \leq 18\tau^2\eta^2\zeta^2 + 4\tau\eta^2\sigma^2,$$

*where $\mathcal{F}^{t,0}$ is the $\sigma$-field representing all the historical information up to the start of the t-th round.*

*Proof of Lemma 6.*

$$\mathbb{E}\left[\left\|\boldsymbol{x}_1^{t,k+1} - \boldsymbol{x}_2^{t,k+1}\right\|^2\Big|\mathcal{F}^{t,k}\right] = \mathbb{E}\left[\left\|\boldsymbol{x}_1^{t,k} - \boldsymbol{x}_2^{t,k} - \eta\left(g_1(\boldsymbol{x}_1^{t,k}) - g_2(\boldsymbol{x}_2^{t,k})\right)\right\|^2\Big|\mathcal{F}^{t,k}\right] \tag{35}$$

$$\leq \left\|\boldsymbol{x}_1^{t,k} - \boldsymbol{x}_2^{t,k}\right\|^2 - 2\eta\left\langle\nabla F_1(\boldsymbol{x}_1^{t,k}) - \nabla F_2(\boldsymbol{x}_2^{t,k}), \boldsymbol{x}_1^{t,k} - \boldsymbol{x}_2^{t,k}\right\rangle$$

$$+ \eta^2\left\|\nabla F_1(\boldsymbol{x}_1^{t,k}) - \nabla F_2(\boldsymbol{x}_2^{t,k})\right\|^2 + 2\eta^2\sigma^2 \tag{36}$$

where the last term in (36) is from Assumption 3 for both $g_1(\boldsymbol{x}_1^{t,k})$ and $g_2(\boldsymbol{x}_2^{t,k})$.

Following Assumption 4, the second term of (36) is bounded as

$$-\left\langle \nabla F_1(\boldsymbol{x}_1^{t,k}) - \nabla F_2(\boldsymbol{x}_2^{t,k}), \boldsymbol{x}_1^{t,k} - \boldsymbol{x}_2^{t,k} \right\rangle \leq -\left\langle \nabla F(\boldsymbol{x}_1^{t,k}) - \nabla F(\boldsymbol{x}_2^{t,k}), \boldsymbol{x}_1^{t,k} - \boldsymbol{x}_2^{t,k} \right\rangle$$
$$+ \left\| \boldsymbol{x}_1^{t,k} - \boldsymbol{x}_2^{t,k} \right\| \left( \left\| \nabla F_1(\boldsymbol{x}_1^{t,k}) - \nabla F(\boldsymbol{x}_1^{t,k}) \right\| + \left\| \nabla F_2(\boldsymbol{x}_2^{t,k}) - \nabla F(\boldsymbol{x}_2^{t,k}) \right\| \right) \tag{37}$$

$$\leq -\left\langle \nabla F(\boldsymbol{x}_1^{t,k}) - \nabla F(\boldsymbol{x}_2^{t,k}), \boldsymbol{x}_1^{t,k} - \boldsymbol{x}_2^{t,k} \right\rangle + 2\zeta \left\| \boldsymbol{x}_1^{t,k} - \boldsymbol{x}_2^{t,k} \right\| \tag{38}$$

$$\leq -\frac{1}{L} \left\| \nabla F(\boldsymbol{x}_1^{t,k}) - \nabla F(\boldsymbol{x}_2^{t,k}) \right\|^2 + 2\zeta \left\| \boldsymbol{x}_1^{t,k} - \boldsymbol{x}_2^{t,k} \right\| \tag{39}$$

$$\leq -\frac{1}{L} \left\| \nabla F(\boldsymbol{x}_1^{t,k}) - \nabla F(\boldsymbol{x}_2^{t,k}) \right\|^2 + \frac{1}{2\eta\tau} \left\| \boldsymbol{x}_1^{t,k} - \boldsymbol{x}_2^{t,k} \right\|^2 + 2\eta\tau\zeta^2 \tag{40}$$

where (39) uses smoothness and convexity properties as per Assumption 2, (40) uses AM-GM inequality (weighted sum version), which is $uv \leq \frac{u^2}{2\epsilon} + \frac{\epsilon v^2}{2}$, with $u = \left\| \boldsymbol{x}_1^{t,k} - \boldsymbol{x}_2^{t,k} \right\|$, $v = 2\zeta$, and $\epsilon = \eta\tau$.

Similarly, the third term of (36) is bounded as

$$\left\| \nabla F_1(\boldsymbol{x}_1^{t,k}) - \nabla F_2(\boldsymbol{x}_2^{t,k}) \right\|^2 \leq \left( 2\zeta + \left\| \nabla F_1(\boldsymbol{x}_1^{t,k}) - \nabla F_2(\boldsymbol{x}_2^{t,k}) \right\| \right)^2 \tag{41}$$

$$= 4\zeta^2 + \left\| \nabla F_1(\boldsymbol{x}_1^{t,k}) - \nabla F_2(\boldsymbol{x}_2^{t,k}) \right\|^2 + 4\zeta \left\| \nabla F_1(\boldsymbol{x}_1^{t,k}) - \nabla F_2(\boldsymbol{x}_2^{t,k}) \right\| \tag{42}$$

$$\leq 3 \left\| \nabla F_1(\boldsymbol{x}_1^{t,k}) - \nabla F_2(\boldsymbol{x}_2^{t,k}) \right\|^2 + 6\zeta^2, \tag{43}$$

where (41) uses Assumption 4, (43) is obtained using AM-GM inequality on the last term of (42), which is $uv \leq \frac{u^2}{2\epsilon} + \frac{\epsilon v^2}{2}$, with $u = \left\| \nabla F_1(\boldsymbol{x}_1^{t,k}) - \nabla F_2(\boldsymbol{x}_2^{t,k}) \right\|$, $v = 4\zeta$, and $\epsilon = \frac{1}{4}$.

Putting all these results together gives us

$$\mathbb{E}\left[ \left\| \boldsymbol{x}_1^{t,k+1} - \boldsymbol{x}_2^{t,k+1} \right\|^2 \Big| \mathcal{F}^{t,k} \right] \leq \left( 1 + \frac{1}{\tau} \right) \left\| \boldsymbol{x}_1^{t,k} - \boldsymbol{x}_2^{t,k} \right\|^2 + 4\tau\eta^2\zeta^2 + 6\eta^2\zeta^2 + 2\eta^2\sigma^2 \tag{44}$$

$$\leq \left( 1 + \frac{1}{\tau} \right) \left\| \boldsymbol{x}_1^{t,k} - \boldsymbol{x}_2^{t,k} \right\|^2 + 10\tau\eta^2\zeta^2 + 2\eta^2\sigma^2. \tag{45}$$

where (44) drops $\left\| \nabla F_1(\boldsymbol{x}_1^{t,k}) - \nabla F_2(\boldsymbol{x}_2^{t,k}) \right\|^2$ term, since the resulting term is always negative given that $\eta \leq \frac{1}{4L}$.

Telescoping gives us

$$\mathbb{E}\left[ \left\| \boldsymbol{x}_1^{t,k} - \boldsymbol{x}_2^{t,k} \right\|^2 \Big| \mathcal{F}^{t,0} \right] \leq \frac{\left( 1 + \frac{1}{\tau} \right)^k - 1}{\frac{1}{\tau}} \cdot \left( 10\tau\eta^2\zeta^2 + 2\eta^2\sigma^2 \right) \tag{46}$$

$$\leq 18\tau^2\eta^2\zeta^2 + 4\tau\eta^2\sigma^2, \tag{47}$$

where the multiplier in (46) is obtained from $\sum_{j=0}^{k-1} \left( 1 + \frac{1}{\tau} \right)^j$ and its numerator is upper bounded by a scalar value of 1.8. Then, by convexity, we have

$$\mathbb{E}\left[ \left\| \boldsymbol{x}_i^{t,k} - \overline{\boldsymbol{x}}^{t,k} \right\|^2 \Big| \mathcal{F}^{t,0} \right] \leq 18\tau^2\eta^2\zeta^2 + 4\tau\eta^2\sigma^2, \quad \forall i \in [N], \tag{48}$$

which completes the proof.

$$\square$$

## A.6. Convergence of the Allocation Algorithm

### A.6.1. ASSUMPTIONS

**Assumption 5** (Finite state space). The state/allocation space $\mathcal{A}$ is finite: $|\mathcal{A}| = M^N < \infty$.

**Assumption 6** (Bounded cost function). The objective function $f : \mathcal{A} \to \mathbb{R}$ is bounded: $\exists \xi > 0$ such that $|f(\boldsymbol{a})| \leq \xi$ for all $\boldsymbol{a} \in \mathcal{A}$.

**Assumption 7** (Irreducibility). For each fixed $T > 0$, the induced Markov chain (with acceptance probabilities depending on $T$) is irreducible on $\mathcal{A}$. That is, for any $\boldsymbol{a}, \boldsymbol{a}' \in \mathcal{A}$, we can reach $\boldsymbol{a}'$ from $\boldsymbol{a}$ with a positive probability in a finite number of steps.

**Assumption 8** (Aperiodicity). For each fixed $T > 0$, the chain is aperiodic: there is no integer $d > 1$ such that transitions occur only in multipliers of $d$. Equivalently, for each $\boldsymbol{a} \in \mathcal{A}, \gcd\{m \mid P_T^m(\boldsymbol{a} \to \boldsymbol{a}) > 0\} = 1$.

**Assumption 9** (Annealing schedule). The temperature $T_k$ satisfies:

1. $T_k \to 0$ as $k \to \infty$.
2. $\sum_{k=1}^{\infty} \exp(-\Delta f / T_k) = \infty$, where $\Delta f = \min_{\boldsymbol{a} \neq \boldsymbol{a}'} |f(\boldsymbol{a}) - f(\boldsymbol{a}')| > 0$ is the smallest nonzero gap of $f$.

A classic example is $T_k = \dfrac{1}{\log(k + k_0)}$ with $k_0 > 1$.

### A.6.2. PROOF OF THEOREM 2

*Proof.* Let $\{A_k\}$ be our Markov chain on $\mathcal{A}$. Denote the transition probability at iteration $k$ by $P_k(\boldsymbol{a} \to \boldsymbol{a}')$. We show that, almost surely, the chain eventually remains (or keeps returning) to a global minimizer.

First, fix a temperature $T > 0$. By Assumptions 7 and 8, the *homogeneous* Markov chain with transitions

$$P_T(\boldsymbol{a} \to \boldsymbol{a}') = \begin{cases} \exp\left(-\dfrac{f(\boldsymbol{a}') - f(\boldsymbol{a})}{T}\right), & f(\boldsymbol{a}') > f(\boldsymbol{a}), \\ 1, & f(\boldsymbol{a}') \leq f(\boldsymbol{a}), \end{cases} \tag{49}$$

is irreducible and aperiodic on the finite state space $\mathcal{A}$. Therefore, it has a unique stationary distribution $\pi_T$.

Standard Metropolis-Hastings arguments show that

$$\pi_T(\boldsymbol{a}) \propto \exp\left(-\frac{f(\boldsymbol{a})}{T}\right). \tag{50}$$

As $T \to 0$, $\exp(-f(\boldsymbol{a})/T)$ is maximized by allocations/states $\boldsymbol{a} \in \mathcal{A}$ that minimize $f$. In fact, if $\boldsymbol{a}^\star$ is a global minimizer (with $f(\boldsymbol{a}^\star) = f_{\min}$), then for any $\boldsymbol{a}$ with $f(\boldsymbol{a}) > f_{\min}$,

$$\frac{\pi_T(\boldsymbol{a}^\star)}{\pi_T(\boldsymbol{a})} = \exp\left(-\frac{f(\boldsymbol{a}^\star) - f(\boldsymbol{a})}{T}\right) \xrightarrow[T \to 0]{} 0. \tag{51}$$

Hence, as $T \to 0$, all stationary mass concentrates on the set of global minima.

However, in our algorithm, $T$ is not fixed but varies with iteration $k$. Thus $\{A_k\}$ is a *time-inhomogeneous* Markov chain whose transition matrix $P_k$ depends on $T_k$. The chain does not, in general, admit a single stationary distribution.

If $\{T_k\}$ decreases slowly enough, the chain nearly equilibrates around each temperature. This ensures we do not remain trapped in a suboptimal local minimum.

Consider a suboptimal state $\boldsymbol{a}$ where $f(\boldsymbol{a}) > f_{\min}$. Because $\mathcal{A}$ is finite, there is a finite path from $\boldsymbol{a}$ to some global minimizer $\boldsymbol{a}^\star$ along which $\max\{f(\cdot)\}$ is well-defined. Let $\Delta(\boldsymbol{a} \to \boldsymbol{a}^\star)$ be the "energy barrier" above $\max\{f(\boldsymbol{a}), f(\boldsymbol{a}^\star)\}$ along that path – i.e., the minimal extra cost one must pay to move from $\boldsymbol{a}$ eventually down to $\boldsymbol{a}^\star$. Formally,

$$\Delta(\boldsymbol{a} \to \boldsymbol{a}^\star) = \min_{\gamma: \boldsymbol{a} \to \boldsymbol{a}^\star} \max_{x \in \gamma} \left[ f(x) - \min\{f(\boldsymbol{a}), f(\boldsymbol{a}^\star)\} \right]. \tag{52}$$

Because $f(\boldsymbol{a}^\star) < f(\boldsymbol{a})$, we have $\Delta(\boldsymbol{a} \to \boldsymbol{a}^\star) > 0$. A single uphill step in cost $\delta$ has acceptance probability of $\exp(-\delta/T_k)$.

**Why we need** $\sum_k \exp\left[-(\Delta f)/T_k\right] = \infty$**:** Let $\Delta_{\max}$ be the maximum barrier needed to reach any global minimizer from any suboptimal state:

$$\Delta_{\max} = \max_{\boldsymbol{a}:f(\boldsymbol{a})>f_{\min},\boldsymbol{a}^\star:f(\boldsymbol{a}^\star)=f_{\min}} \Delta(\boldsymbol{a} \rightarrow \boldsymbol{a}^\star). \tag{53}$$

Then any upward move $\delta \leq \Delta_{\max}$ is accepted with probability at least $\exp\left(-\Delta_{\max}/T_k\right)$. If

$$\sum_{k=1}^{\infty} \exp\left(-\frac{\Delta_{\max}}{T_k}\right) = \infty, \tag{54}$$

we get infinitely many chances (with positive probability) to surmount each barrier. By the Borel-Cantelli lemma, almost surely, the chain eventually does surmount every finite barrier and thus can move from any suboptimal state to a strictly better region. Repeatedly, the chain escapes local minima with probability 1.

Once the chain hits a global minimum $\boldsymbol{a}^\star$ at sufficiently small $T_k$, any transition to a higher-cost state is exponentially unlikely (with probability $\exp\left(-\left[f(\boldsymbol{a}') - f(\boldsymbol{a}^\star)\right]/T_k\right)$). As $T_k \rightarrow 0$, these moves become negligible, causing the chain to remain in (or return quickly to) a global minimizer. Consequently,

$$\lim_{k\to\infty} P(A_k \in \{\boldsymbol{a} : f(\boldsymbol{a}) = f_{\min}\}) = 1, \tag{55}$$

proving convergence to a global minimizer with probability 1.

This completes the proof.

$\square$

## B. Implementation Details

**Evaluation metrics.** We evaluate our approach and the baseline methods in terms of both predictive performance and fairness/incentivization. For predictive performance, we use balanced accuracy. Fairness, on the other hand, can be assessed using several metrics, including the incentivized participation rate (IPR) proposed by (Cho et al., 2022), the Pearson correlation coefficient and the collaboration gain spread (CGS) proposed by (Tastan et al., 2025). However, we exclude IPR from our analysis, as our allocation algorithm inherently guarantees a perfect IPR score by design. Instead, we primarily benchmark our results using Pearson correlation and CGS.

**Implementation details.** We use cross-entropy loss for all image and language classification tasks and maintain consistent training hyperparameters across all experiments. The optimizer of choice is SGD with momentum, with a default initial learning rate of $0.01$. A learning rate scheduler is applied, reducing the learning rate by a factor of $0.1$ at rounds $50$ and $75$, when the total number of communication rounds is set to $100$. The total number of communications is set as follows:

- CIFAR-10, CIFAR-100, and SST: $T = 100$,

- MNIST, FMNIST, and SVHN: $T = 50$.

In each round, clients perform one local epoch of training. The batch size is fixed at $128$ across all experiments. Additionally, we specify one parameter: the minimum width of the slimmable network, which is set to $p_{\min} = 0.25$ unless stated otherwise. The maximum width is always kept at $p_{\max} = 1.0$. All experiments were carried out on NVIDIA A100-SXM4-40GB GPUs, with each run utilizing a single GPU.

**Architecture details.** To ensure the reproducibility of our experiments, we used easily implementable model architectures. Specifically, we employed four different architectures:

1. Convolutional neural networks (CNN) – a lightweight CNN with one convolutional layer followed by two fully connected layers, totaling 0.0606M parameters. This model is used for the MNIST and FMNIST datasets.

---

**Algorithm 2** Aequa (with training-time model rewards)

---

**Input:** minimum width $p_{\min}$, maximum width $p_{\max}$, number of communication rounds $T$, number of participants $N$, randomly initialized weights $x^0$, number of local iterations $E$, momentum factor $\gamma = 0.5$, CA - contribution assessment algorithm (e.g. CGSV, ShapFed, FedFAIM, etc.)

1: **for** each round $t \leftarrow 0, 1, \ldots, T$ **do**          ▷ *Communication rounds*
2:     **if** $t = 0$ **then**
3:        $p_{\max}^{(i,t)} \leftarrow p_{\max}, \forall i \in [N]$
4:        Server broadcasts the full-sized model ($p_{\max}$-model) to each client $i \in [N]$
5:     **else**
6:        Server broadcasts $p_{\max}^{(i,t)}$-submodel to each client $i \in [N]$
7:     **end if**
8:     **for** each participant $i \in [N]$ **do**
9:        **for** $k \leftarrow 0, 1, \ldots, E$ **do**          ▷ *Local iterations*
10:          Sample width $p_{(i,k)} \leftarrow \mathcal{U}(p_{\min}, p_{\max}^{(i,t)})$ uniformly
11:          Update the weights of the model corresponding to $p_{\max}^{(i,t)}$ and $p_{(i,k)}$ widths
12:          Send the updated weights of the $p_{\max}^{(i,t)}$-submodel
13:        **end for**
14:     **end for**
15:     Server computes the contribution of each client $\tilde{c}_i$ using gradients according to $p_{\min}$-submodel via the CA algorithm
16:     Server updates the contribution of each client using Eq. 4, $c_i^t \leftarrow \gamma c_i^{t-1} + (1 - \gamma)\tilde{c}_i$ if $t > 0$ else $c_i^t \leftarrow \tilde{c}_i$
17:     Server updates the participant widths using the reward mechanism in Eq. 5          ▷ *Any reward mechanism*
18:     Server updates $x^{t+1}$ using the masked averaging (Mohtashami et al., 2022; Tastan et al., 2024b)
19: **end for**

---

2. Enhanced CNN model – A more complex architecture with two convolutional layers followed by two fully connected layers, comprising 2.0566M parameters. This model is used for the SVHN dataset.

3. ResNet18 (He et al., 2016) – a deeper model with 10.7527M parameters, used for the CIFAR-10 and CIFAR-100 datasets.

4. Long short-term memory network (LSTM) – this model includes an embedding layer of dimension 300, an LSTM layer, and three fully connected layers, comprising 6.1461M parameters. It is used for the SST dataset.

For slimmable model implementation, we apply width slimming to the following layers: convolutional layers, fully connected (FC) layers, batch normalization (BatchNorm), and LSTM layers. Slimming convolutional, fully connected, and LSTM layers is straightforward; however, BatchNorm layers require special treatment due to the inconsistencies between training and testing. To address this, we implement switchable batch normalization, which maintains multiple sets of batch normalization statistics corresponding to different model widths. For further details, refer to (Yu et al., 2019).

**Baseline approaches.** We describe the baseline methods used for comparison.

**FedAvg-FT.** The standard FedAvg algorithm (McMahan et al., 2017) returns the same model to all clients at each FL iteration, making the computation of the Pearson correlation coefficient undefined. To address this, following IAFL (Wu et al., 2024), we introduce a fine-tuning step at the end of collaboration, where each client trains the global model locally for an additional round without sharing the updates with the server. This approach enables each client to receive a personalized model, allowing for a valid estimation of the Pearson correlation coefficient.

**CGSV.** In cosine-gradient Shapley-value (CGSV) (Xu et al., 2021), the server estimates each participant's contribution using gradient alignment, computed as the cosine similarity between an individual gradient and the aggregated gradient. During the broadcasting phase, updates are sparsified based on these estimated contributions before being sent back to clients.

**IAFL.** For incentive-aware federated learning (IAFL), we adopt the same setting described in (Wu et al., 2024) under the most fair scenario, which corresponds to setting the hyperparameters $\kappa = 0$ (sharing parameter) and $q = 0$ (stochastic recovery probability). Additionally, we use standalone accuracies as the contribution measure, which is consistently used in (Wu et al., 2024).

**ShapFed.** We incorporate Shapley-driven federated learning (ShapFed) (Tastan et al., 2024a) and CGSV as a contribution assessment algorithms within our method. ShapFed estimates contributions using last-layer gradients instead of full-model parameters, making it an efficient and accurate approach in certain scenarios. To better align with our objective, we modify the ShapFed algorithm to use the last $m$ layers instead of only the classification layer, as our method does not require class-specific contribution values. We set $m = 10$ in CIFAR-10 experiment with ResNet-18 architecture.

## C. Extension to Training-time Model Rewards

In this section, we provide Algorithm 2, which extends our proposed Aequa framework to incorporate training-time model rewards. While our primary allocation mechanism focuses on post-training model distribution, this extension enables dynamic model adaptation during training, where clients receive real-time adjustments to their assigned model widths based on their contributions. However, our approach remains flexible and can seamlessly integrate with the primary allocation mechanism. We showcase this extension to demonstrate that our method is compatible with various reward allocation algorithms, emphasizing its adaptability.

Algorithm 2 outlines the federated training process with contribution-aware model scaling. The algorithm initializes all clients with the full-sized model. As training progresses, client contributions are continuously assessed using a contribution assessment (CA) algorithm, such as CGSV (Xu et al., 2021), ShapFed (Tastan et al., 2024a), or FedFAIM (Shi et al., 2022). These contributions are then updated iteratively using a momentum-based update rule, ensuring a fair and stable estimation over multiple communication rounds.

Based on the updated contributions, the server dynamically adjusts the model width allocated to each client, applying a reward mechanism that incentivizes higher-performing participants. This ensures that clients who contribute more to the global model benefit from larger subnetworks, while maintaining fairness in model distribution. The global model update is performed using masked averaging (Mohtashami et al., 2022; Tastan et al., 2024b), a robust aggregation technique that ensures stability across varying model widths. The full details of Algorithm 2 are presented below.

## D. Additional Experiments

### D.1. Performance vs model width

**Remaining results of Figure 1.** In this section, we present the remaining plots corresponding to Figure 1 from the main paper, which analyzed CIFAR-10 and CIFAR-100 on the ResNet-18 architecture. Here, we extend the analysis by including Figure 6, which illustrates the same performance vs. model width relationship for four additional datasets: MNIST, FMNIST, SVHN, and SST and under a homogeneous partitioning strategy.

As described in Appendix B, these datasets are trained using different model architectures: MNIST and FMNIST utilize a lightweight CNN, SVHN is trained on a more complex CNN, and SST is trained using an LSTM network. Despite the differences in model complexity, the width-accuracy relationship remains monotonic, demonstrating that even small-sized network architectures exhibit a clear relation between model width and predictive performance.

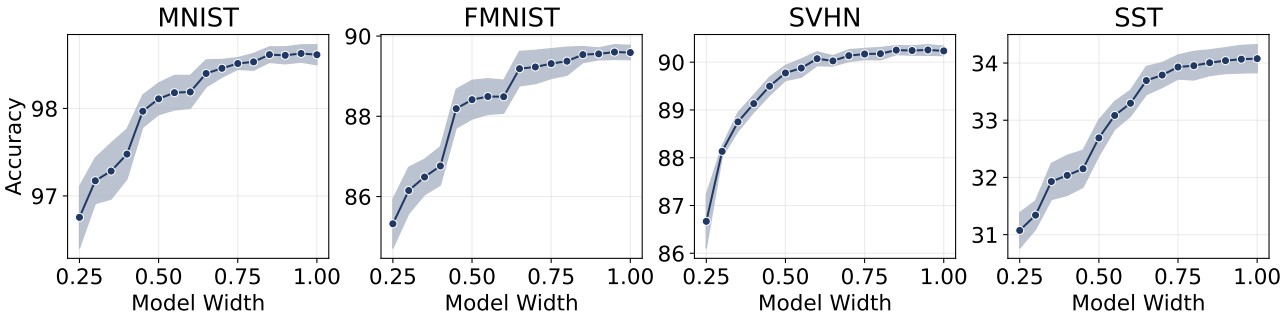

Figure 6: Performance vs. network width ($p \in [p_{\min} = 0.25, p_{\max} = 1.0]$) on MNIST, FMNIST, SVHN, and SST datasets using CNN and LSTM models. This figure extends the results presented in Figure 1.

An important observation is that as the model complexity increases, the range between minimum and maximum accuracy widens, making the trade-off between model width and accuracy more pronounced.

### D.2. Predictive performance

**Remaining results of Table 1.** For completeness, we present the full results of our approach alongside the baseline methods across all partitioning strategies. Table 1 previously reported results for a subset of partitions, while the complete results are provided in Table 5.

From the overall results presented in Table 5, it is evident that our method outperforms all other approaches, including FedAvg, and significantly surpasses fairness-based methods. Specifically, across 9 partitioning strategies and 6 datasets per partition, our approach achieves the best performance in 36 cases. In the remaining 18 cases, it ranks as the second-best, performing on par with the FedAvg algorithm.

### D.3. Pearson correlation

**Remaining results of Table 2.** In Table 6, we present the complete results of the experiments on Pearson correlation, extending the findings of Table 2, which reported results for only a subset of partitions. The results demonstrate that our algorithm outperforms all other methods in all $54$ cases.

As discussed in the fairness analysis (Section 5.2), our method consistently achieves near-perfect correlation coefficients, typically in the range of $0.98$ to $0.99$. While the theoretical analysis guarantees a perfect correlation, in practice, the continuity assumption slightly violated due to the need for discretizing the accuracy range $[\ell, u]$.

Additionally, in experiments using the ResNet-18 architecture, we introduce discretization by modifying the uniform sampling strategy. Instead of continuous sampling, we define a set of model widths sampled from a bucket of predefined values, starting from $0.25, 0.3, 0.35$, and increasing in increments of $0.05$, up to $1.0$.

### D.4. MCG and CGS results

In this section, we provide the detailed results for mean collaboration gain (MCG) and collaboration gain spread (CGS), expanding upon the summary presented in Section 6.3. Table 7 presents a comprehensive comparison of Aequa against baseline methods across all experimental settings. The results confirm that Aequa consistently achieves superior performance, outperforming other methods in $42$ out of $54$ cases.

For cases where Aequa does not achieve the lowest CGS, we highlight the MCG values in green, demonstrating that Aequa still significantly outperforms other methods in MCG (by $\sim \times 3$). This suggests that even when another method achieves a comparable CGS, it does so at the expense of lower mean collaboration gain, indicating a weaker overall incentivization effect. Such cases occur in extreme heterogeneous settings.

As explained earlier, another scenario where Aequa underperforms compared to IAFL is in the quantity skew setting. This performance gap is primarily due to the selected value of $p_{\min}$, which determines the minimum accuracy $\ell$ assigned to low-contributing clients. To examine the impact of this parameter, we conduct an additional set of experiments where $p_{\min}$ is adjusted to $0.1$, specifically for the quantity skew partition. The results, presented in Table 8, show that under this revised configuration, Aequa outperforms other baselines methods in $11$ out of $12$ cases, bringing the overall success rate of $48$ out of $54$ cases.

The partitioning strategies where Aequa consistently outperforms all methods include homogeneous, Dirichlet ($\alpha = \{1.0, 2.0, 5.0\}$), quantity skew $(0.15, 6)$, and label skew $(C = 3)$. In contrast, CGSV performs well in extremely heterogeneous settings, achieving positive MCG values, but it tends to yield negative values in other settings, limiting its generalizability. FedAvg-FT, on the other hand, consistently achieves positive MCG values but exhibits higher CGS values, indicating a lack of fairness.

### D.5. Per-participant performance

In this section, we present a per-participant performance analysis of Aequa and other baseline methods on the CIFAR-100 dataset under the following partitioning strategies: Dirichlet ($\alpha = 0.5$), label skew $(0.15, 6)$ and quantity skew $(0.4, 2)$.

From the results, we observe that IAFL exhibits high variability in the first two partitioning strategies, and in some random

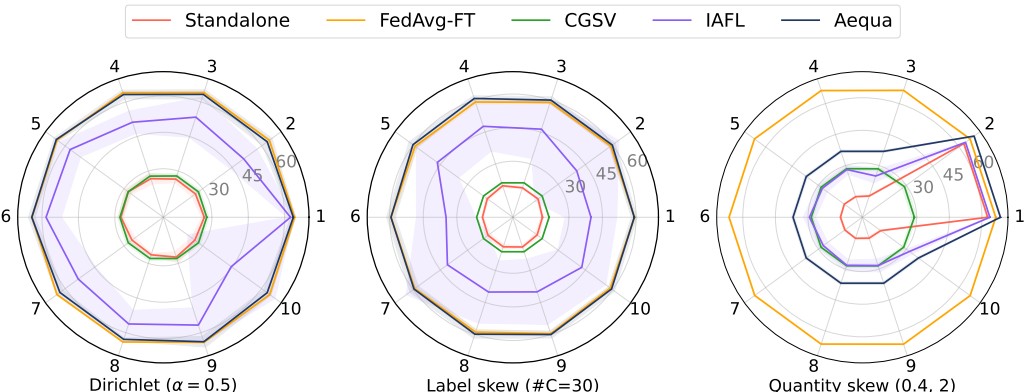

Figure 7: Per-participant performance comparison of Aequa and other baseline methods on CIFAR-100 dataset under Dirichlet ($\alpha = 0.5$), label skew $(0.15, 6)$ and quantity skew $(0.4, 2)$ partitions.

seeds, IAFL fails to achieve a positive collaboration gain. On the other hand, CGSV remains very close to standalone accuracy, significantly limiting the overall collaboration gain of the participants. Meanwhile, Aequa achieves performance comparable to FedAvg-FT, while precisely capturing the correlation between final model accuracies and client contributions, reinforcing its fairness properties.

In the quantity skew partition, we observe that Clients 1 and 2 are the highest contributors, as each holds 40% of the dataset. The goal of a fair allocation algorithm is to match this distribution while expanding the total collaboration gain. Both IAFL and Aequa produce similar-shaped performance plots to the standalone accuracy, consistently ensuring positive collaboration gains. However, Aequa provides even better performance to Clients 1 and 2 compared to FedAvg-FT, while also maximizing the mean collaboration gain (MCG) across all participants, demonstrating its effectiveness in fair model allocation.

### D.6. Model width as a reward

In this section, we explore the transferability of Aequa when the server lacks prior knowledge of each sub-model's performance (does not possess a validation set). In this scenario, participants are rewarded directly based on the model width ($\boldsymbol{p}$), rather than using explicit performance-based allocations ($\boldsymbol{a}$).

To implement this approach, we use standalone accuracies as the contribution measure, and the allocation problem becomes mapping model widths to participant contributions. We employ the same allocation algorithm described in Section 4.2, but modify the utility measure to be computed using normalized contributions, i.e. $c_i / \max_k c_k$, ensuring that both vectors share the same range of values before executing the allocation process.

Table 9 presents the results of this experiment, covering three datasets across all partitioning strategies. The results demonstrate that the correlation coefficient between participant contributions (input) and assigned model widths (output) is perfectly aligned, consistently achieving a correlation coefficient of 1.0.

Additionally, we report the correlation coefficient between the contribution measure and the assigned sub-model's corresponding accuracy. The findings indicate that even when the server lacks knowledge of each sub-model's accuracy, high fairness performance is still achievable. Compared to the results presented in the main paper, this method still outperforms all baseline approaches, demonstrating that Aequa maintains strong fairness properties.

Table 5: The predictive performance of our method and other baselines using different dataset partitions. The results are averaged over five independent evaluations.

| PARTITION | DATASET | FEDAVG | CGSV | IAFL | AEQUA |
|---|---|---|---|---|---|
| HOMOGENEOUS | MNIST | **98.67** ± 0.07 | 90.62 ± 2.32 | 98.40 ± 0.13 | 98.60 ± 0.11 |
| | FMNIST | 89.45 ± 0.33 | 77.01 ± 2.51 | 88.54 ± 0.25 | **89.63** ± 0.19 |
| | SVHN | **90.54** ± 0.18 | 77.61 ± 3.16 | 89.64 ± 0.13 | 90.18 ± 0.15 |
| | CIFAR-10 | 89.99 ± 0.23 | 61.29 ± 2.92 | 88.42 ± 0.07 | **90.84** ± 0.26 |
| | CIFAR-100 | 65.92 ± 0.22 | 35.36 ± 0.77 | 63.23 ± 0.30 | **67.83** ± 0.32 |
| | SST | 34.44 ± 1.33 | 30.12 ± 1.03 | 34.02 ± 0.51 | **34.44** ± 1.19 |
| HETEROGENEOUS: DIRICHLET ($\alpha = 0.1$) | MNIST | **97.38** ± 0.62 | 94.06 ± 1.79 | 88.45 ± 7.87 | 97.30 ± 0.58 |
| | FMNIST | 83.32 ± 1.78 | 71.51 ± 8.20 | 66.46 ± 4.72 | **84.60** ± 1.32 |
| | SVHN | **86.38** ± 0.87 | 72.48 ± 4.81 | 68.71 ± 7.85 | 86.33 ± 1.00 |
| | CIFAR-10 | 74.73 ± 3.65 | 48.77 ± 5.02 | 46.36 ± 9.31 | **75.97** ± 3.36 |
| | CIFAR-100 | 61.16 ± 0.25 | 34.16 ± 1.63 | 43.38 ± 4.53 | **63.42** ± 0.54 |
| | SST | 32.17 ± 1.60 | 21.54 ± 1.89 | 27.28 ± 3.06 | **33.54** ± 1.48 |
| HETEROGENEOUS: DIRICHLET ($\alpha = 0.5$) | MNIST | **98.45** ± 0.13 | 93.18 ± 1.70 | 97.69 ± 0.44 | 98.29 ± 0.16 |
| | FMNIST | 87.86 ± 0.42 | 79.24 ± 3.28 | 85.42 ± 1.67 | **88.20** ± 0.49 |
| | SVHN | **89.33** ± 0.25 | 80.20 ± 2.10 | 87.08 ± 0.79 | 89.14 ± 0.28 |
| | CIFAR-10 | 87.74 ± 0.23 | 65.12 ± 1.82 | 81.99 ± 2.46 | **88.75** ± 0.39 |
| | CIFAR-100 | 64.54 ± 0.30 | 32.19 ± 0.96 | 62.37 ± 1.35 | **66.39** ± 0.41 |
| | SST | 33.52 ± 0.80 | 23.82 ± 2.03 | 29.64 ± 2.15 | **34.21** ± 1.08 |
| HETEROGENEOUS: DIRICHLET ($\alpha = 1.0$) | MNIST | **98.52** ± 0.14 | 92.71 ± 1.53 | 98.17 ± 0.15 | 98.48 ± 0.13 |
| | FMNIST | 88.72 ± 0.36 | 78.76 ± 1.40 | 87.67 ± 0.22 | **89.09** ± 0.23 |
| | SVHN | **89.96** ± 0.30 | 79.53 ± 2.50 | 88.51 ± 0.26 | 89.75 ± 0.22 |
| | CIFAR-10 | 88.76 ± 0.26 | 65.88 ± 5.02 | 84.02 ± 1.12 | **89.67** ± 0.28 |
| | CIFAR-100 | 65.06 ± 0.29 | 33.92 ± 1.32 | 64.37 ± 0.68 | **67.06** ± 0.44 |
| | SST | 34.00 ± 0.87 | 25.24 ± 2.55 | 31.26 ± 1.22 | **34.15** ± 1.07 |
| HETEROGENEOUS: DIRICHLET ($\alpha = 2.0$) | MNIST | **98.58** ± 0.15 | 92.56 ± 1.54 | 98.22 ± 0.13 | 98.50 ± 0.09 |
| | FMNIST | 89.12 ± 0.36 | 79.42 ± 2.17 | 88.27 ± 0.28 | **89.46** ± 0.18 |
| | SVHN | **90.19** ± 0.10 | 78.16 ± 2.46 | 88.91 ± 0.14 | 89.94 ± 0.13 |
| | CIFAR-10 | 89.35 ± 0.28 | 68.60 ± 3.36 | 88.09 ± 0.33 | **90.30** ± 0.18 |
| | CIFAR-100 | 65.56 ± 0.22 | 36.14 ± 1.66 | 64.71 ± 0.70 | **67.51** ± 0.29 |
| | SST | 33.45 ± 1.32 | 27.55 ± 0.62 | 33.25 ± 0.27 | **34.58** ± 0.49 |
| HETEROGENEOUS: DIRICHLET ($\alpha = 5.0$) | MNIST | **98.64** ± 0.12 | 91.73 ± 2.95 | 98.31 ± 0.13 | 98.54 ± 0.11 |
| | FMNIST | 89.35 ± 0.31 | 77.03 ± 2.66 | 88.44 ± 0.24 | **89.60** ± 0.15 |
| | SVHN | **90.48** ± 0.16 | 80.52 ± 1.94 | 89.32 ± 0.15 | 90.08 ± 0.29 |
| | CIFAR-10 | 89.56 ± 0.29 | 68.09 ± 5.05 | 88.47 ± 0.50 | **90.35** ± 0.14 |
| | CIFAR-100 | 65.79 ± 0.29 | 37.28 ± 0.66 | 65.61 ± 0.39 | **67.77** ± 0.16 |
| | SST | 34.26 ± 1.01 | 28.75 ± 0.31 | 33.41 ± 1.17 | **34.47** ± 0.55 |
| QUANTITY SKEW: IMBALANCED $(0.15, 6)$ | MNIST | **98.69** ± 0.10 | 93.22 ± 0.99 | 98.40 ± 0.12 | 98.62 ± 0.09 |
| | FMNIST | 89.53 ± 0.24 | 78.73 ± 2.14 | 88.53 ± 0.26 | **89.72** ± 0.17 |
| | SVHN | **90.59** ± 0.17 | 77.54 ± 2.34 | 89.51 ± 0.13 | 90.26 ± 0.16 |
| | CIFAR-10 | 90.00 ± 0.13 | 66.89 ± 1.85 | 89.51 ± 0.15 | **90.71** ± 0.14 |
| | CIFAR-100 | 65.88 ± 0.38 | 39.62 ± 0.74 | 64.60 ± 0.25 | **68.24** ± 0.11 |
| | SST | 34.26 ± 0.98 | 29.53 ± 1.02 | 33.70 ± 0.70 | **34.64** ± 1.01 |
| QUANTITY SKEW: IMBALANCED $(0.4, 2)$ | MNIST | **98.69** ± 0.09 | 92.73 ± 1.34 | 98.42 ± 0.16 | 98.63 ± 0.12 |
| | FMNIST | 89.58 ± 0.29 | 76.14 ± 4.79 | 88.51 ± 0.25 | **89.78** ± 0.19 |
| | SVHN | **90.69** ± 0.15 | 79.11 ± 1.35 | 89.49 ± 0.07 | 90.35 ± 0.17 |
| | CIFAR-10 | 89.78 ± 0.14 | 69.13 ± 3.26 | 88.50 ± 0.13 | **90.86** ± 0.15 |
| | CIFAR-100 | 66.30 ± 0.23 | 38.35 ± 1.01 | 63.67 ± 0.10 | **68.80** ± 0.16 |
| | SST | **35.45** ± 1.11 | 26.10 ± 0.65 | 34.65 ± 0.56 | 35.29 ± 0.76 |
| LABEL SKEW: #OC={3, 30} | MNIST | 94.37 ± 3.43 | 79.19 ± 7.94 | 73.10 ± 15.00 | **95.37** ± 1.15 |
| | FMNIST | 79.73 ± 3.80 | 61.54 ± 8.03 | 60.10 ± 8.03 | **80.51** ± 3.27 |
| | SVHN | 79.73 ± 5.89 | 64.07 ± 7.65 | 55.83 ± 11.89 | **80.69** ± 6.05 |
| | CIFAR-10 | 71.88 ± 3.28 | 48.02 ± 3.88 | 44.12 ± 21.15 | **72.40** ± 3.17 |
| | CIFAR-100 | 60.95 ± 1.18 | 35.09 ± 0.42 | 55.26 ± 3.85 | **62.84** ± 1.18 |
| | SST | **33.96** ± 0.35 | 24.88 ± 2.04 | 30.33 ± 1.68 | 33.01 ± 0.90 |
| NUMBER OF TIMES THAT PERFORMS THE BEST | | 18/54 | 0/54 | 0/54 | **36/54** |

Table 6: The incentivization performance of our method and other baselines under different dataset partitions, measured using the Pearson correlation coefficient between the final model accuracies and standalone accuracies. The results are averaged over five independent evaluations.

| PARTITION | DATASET | FEDAVG-FT | CGSV | IAFL | AEQUA |
|---|---|---|---|---|---|
| HOMOGENEOUS | MNIST | $0.07 \pm 0.24$ | $-0.30 \pm 0.16$ | $0.16 \pm 0.20$ | $\mathbf{0.97 \pm 0.02}$ |
| | FMNIST | $0.21 \pm 0.07$ | $-0.09 \pm 0.35$ | $0.27 \pm 0.24$ | $\mathbf{0.98 \pm 0.02}$ |
| | SVHN | $-0.12 \pm 0.36$ | $0.02 \pm 0.17$ | $0.07 \pm 0.28$ | $\mathbf{0.98 \pm 0.02}$ |
| | CIFAR-10 | $0.04 \pm 0.27$ | $0.05 \pm 0.42$ | $-0.01 \pm 0.15$ | $\mathbf{0.99 \pm 0.01}$ |
| | CIFAR-100 | $-0.07 \pm 0.37$ | $-0.19 \pm 0.36$ | $0.02 \pm 0.31$ | $\mathbf{0.96 \pm 0.01}$ |
| | SST | $0.06 \pm 0.25$ | $-0.07 \pm 0.32$ | $-0.03 \pm 0.26$ | $\mathbf{0.98 \pm 0.01}$ |
| HETEROGENEOUS: DIRICHLET ($\alpha = 0.1$) | MNIST | $0.39 \pm 0.37$ | $0.56 \pm 0.18$ | $0.61 \pm 0.23$ | $\mathbf{0.85 \pm 0.03}$ |
| | FMNIST | $0.21 \pm 0.41$ | $0.56 \pm 0.28$ | $0.61 \pm 0.25$ | $\mathbf{0.89 \pm 0.06}$ |
| | SVHN | $0.66 \pm 0.24$ | $0.21 \pm 0.46$ | $0.80 \pm 0.18$ | $\mathbf{0.92 \pm 0.02}$ |
| | CIFAR-10 | $-0.18 \pm 0.35$ | $0.67 \pm 0.14$ | $0.84 \pm 0.15$ | $\mathbf{0.94 \pm 0.02}$ |
| | CIFAR-100 | $-0.17 \pm 0.46$ | $0.30 \pm 0.52$ | $0.89 \pm 0.07$ | $\mathbf{0.99 \pm 0.01}$ |
| | SST | $-0.20 \pm 0.35$ | $0.14 \pm 0.50$ | $0.91 \pm 0.09$ | $\mathbf{0.98 \pm 0.01}$ |
| HETEROGENEOUS: DIRICHLET ($\alpha = 0.5$) | MNIST | $0.32 \pm 0.40$ | $-0.22 \pm 0.48$ | $0.67 \pm 0.16$ | $\mathbf{0.87 \pm 0.03}$ |
| | FMNIST | $0.67 \pm 0.19$ | $0.34 \pm 0.48$ | $0.50 \pm 0.10$ | $\mathbf{0.88 \pm 0.05}$ |
| | SVHN | $0.80 \pm 0.09$ | $-0.18 \pm 0.54$ | $0.59 \pm 0.41$ | $\mathbf{0.90 \pm 0.04}$ |
| | CIFAR-10 | $-0.16 \pm 0.21$ | $0.44 \pm 0.38$ | $0.93 \pm 0.03$ | $\mathbf{0.94 \pm 0.03}$ |
| | CIFAR-100 | $0.19 \pm 0.29$ | $0.38 \pm 0.14$ | $0.58 \pm 0.23$ | $\mathbf{0.99 \pm 0.00}$ |
| | SST | $-0.15 \pm 0.07$ | $-0.10 \pm 0.34$ | $0.88 \pm 0.05$ | $\mathbf{0.94 \pm 0.04}$ |
| HETEROGENEOUS: DIRICHLET ($\alpha = 1.0$) | MNIST | $0.55 \pm 0.20$ | $0.13 \pm 0.48$ | $0.48 \pm 0.40$ | $\mathbf{0.91 \pm 0.03}$ |
| | FMNIST | $0.42 \pm 0.51$ | $0.27 \pm 0.30$ | $0.46 \pm 0.37$ | $\mathbf{0.95 \pm 0.03}$ |
| | SVHN | $0.52 \pm 0.28$ | $0.08 \pm 0.19$ | $0.69 \pm 0.08$ | $\mathbf{0.94 \pm 0.03}$ |
| | CIFAR-10 | $-0.05 \pm 0.15$ | $0.50 \pm 0.41$ | $0.89 \pm 0.04$ | $\mathbf{0.94 \pm 0.02}$ |
| | CIFAR-100 | $-0.08 \pm 0.35$ | $-0.23 \pm 0.51$ | $0.65 \pm 0.37$ | $\mathbf{0.99 \pm 0.00}$ |
| | SST | $0.09 \pm 0.29$ | $-0.16 \pm 0.33$ | $0.71 \pm 0.12$ | $\mathbf{0.94 \pm 0.04}$ |
| HETEROGENEOUS: DIRICHLET ($\alpha = 2.0$) | MNIST | $0.49 \pm 0.32$ | $-0.07 \pm 0.57$ | $0.37 \pm 0.41$ | $\mathbf{0.96 \pm 0.03}$ |
| | FMNIST | $0.50 \pm 0.24$ | $0.03 \pm 0.52$ | $0.51 \pm 0.27$ | $\mathbf{0.96 \pm 0.03}$ |
| | SVHN | $0.67 \pm 0.10$ | $0.09 \pm 0.06$ | $0.72 \pm 0.16$ | $\mathbf{0.95 \pm 0.02}$ |
| | CIFAR-10 | $0.17 \pm 0.28$ | $0.58 \pm 0.21$ | $0.83 \pm 0.15$ | $\mathbf{0.97 \pm 0.03}$ |
| | CIFAR-100 | $-0.26 \pm 0.25$ | $0.05 \pm 0.46$ | $0.65 \pm 0.26$ | $\mathbf{0.99 \pm 0.00}$ |
| | SST | $-0.19 \pm 0.36$ | $0.08 \pm 0.36$ | $0.59 \pm 0.28$ | $\mathbf{0.95 \pm 0.04}$ |
| HETEROGENEOUS: DIRICHLET ($\alpha = 5.0$) | MNIST | $0.26 \pm 0.32$ | $-0.10 \pm 0.40$ | $0.39 \pm 0.46$ | $\mathbf{0.96 \pm 0.03}$ |
| | FMNIST | $0.26 \pm 0.32$ | $-0.32 \pm 0.22$ | $0.12 \pm 0.29$ | $\mathbf{0.97 \pm 0.02}$ |
| | SVHN | $0.09 \pm 0.18$ | $0.14 \pm 0.37$ | $0.62 \pm 0.05$ | $\mathbf{0.98 \pm 0.01}$ |
| | CIFAR-10 | $0.01 \pm 0.16$ | $0.22 \pm 0.54$ | $0.80 \pm 0.08$ | $\mathbf{0.98 \pm 0.02}$ |
| | CIFAR-100 | $0.11 \pm 0.25$ | $0.04 \pm 0.26$ | $0.18 \pm 0.50$ | $\mathbf{0.99 \pm 0.01}$ |
| | SST | $-0.04 \pm 0.21$ | $0.21 \pm 0.32$ | $0.37 \pm 0.36$ | $\mathbf{0.98 \pm 0.01}$ |
| QUANTITY SKEW: IMBALANCED $(0.15, 6)$ | MNIST | $-0.63 \pm 0.18$ | $0.34 \pm 0.80$ | $0.95 \pm 0.04$ | $\mathbf{0.98 \pm 0.01}$ |
| | FMNIST | $-0.45 \pm 0.28$ | $0.49 \pm 0.71$ | $0.93 \pm 0.02$ | $\mathbf{0.98 \pm 0.01}$ |
| | SVHN | $-0.76 \pm 0.12$ | $0.42 \pm 0.75$ | $0.99 \pm 0.01$ | $\mathbf{1.00 \pm 0.00}$ |
| | CIFAR-10 | $-0.37 \pm 0.16$ | $0.98 \pm 0.02$ | $0.99 \pm 0.00$ | $\mathbf{1.00 \pm 0.00}$ |
| | CIFAR-100 | $0.06 \pm 0.40$ | $0.97 \pm 0.03$ | $\mathbf{1.00 \pm 0.00}$ | $\mathbf{1.00 \pm 0.00}$ |
| | SST | $-0.07 \pm 0.48$ | $-0.23 \pm 0.54$ | $0.90 \pm 0.02$ | $\mathbf{0.94 \pm 0.04}$ |
| QUANTITY SKEW: IMBALANCED $(0.4, 2)$ | MNIST | $-0.61 \pm 0.13$ | $-0.33 \pm 0.76$ | $0.76 \pm 0.04$ | $\mathbf{0.96 \pm 0.03}$ |
| | FMNIST | $-0.21 \pm 0.26$ | $0.51 \pm 0.38$ | $0.90 \pm 0.04$ | $\mathbf{0.99 \pm 0.01}$ |
| | SVHN | $-0.54 \pm 0.20$ | $-0.06 \pm 0.79$ | $0.93 \pm 0.01$ | $\mathbf{0.98 \pm 0.01}$ |
| | CIFAR-10 | $-0.41 \pm 0.35$ | $0.20 \pm 0.96$ | $0.92 \pm 0.04$ | $\mathbf{1.00 \pm 0.00}$ |
| | CIFAR-100 | $0.05 \pm 0.25$ | $0.47 \pm 0.75$ | $0.98 \pm 0.01$ | $\mathbf{1.00 \pm 0.00}$ |
| | SST | $-0.05 \pm 0.61$ | $-0.97 \pm 0.01$ | $0.90 \pm 0.07$ | $\mathbf{0.97 \pm 0.01}$ |
| LABEL SKEW: #OC=$\{3, 30\}$ | MNIST | $0.03 \pm 0.41$ | $-0.27 \pm 0.27$ | $0.23 \pm 0.24$ | $\mathbf{0.81 \pm 0.13}$ |
| | FMNIST | $-0.44 \pm 0.27$ | $0.11 \pm 0.45$ | $0.08 \pm 0.26$ | $\mathbf{0.99 \pm 0.01}$ |
| | SVHN | $0.43 \pm 0.25$ | $-0.43 \pm 0.42$ | $0.01 \pm 0.25$ | $\mathbf{0.98 \pm 0.00}$ |
| | CIFAR-10 | $0.19 \pm 0.32$ | $0.12 \pm 0.32$ | $0.22 \pm 0.38$ | $\mathbf{0.97 \pm 0.02}$ |
| | CIFAR-100 | $-0.38 \pm 0.22$ | $0.00 \pm 0.24$ | $0.31 \pm 0.22$ | $\mathbf{0.98 \pm 0.02}$ |
| | SST | $0.45 \pm 0.38$ | $-0.19 \pm 0.48$ | $0.48 \pm 0.37$ | $\mathbf{0.97 \pm 0.03}$ |
| NUMBER OF TIMES THAT PERFORMS THE BEST | | $0/54$ | $0/54$ | $1/54$ | $\mathbf{54/54}$ |

Table 7: The average collaboration gain (MCG) ($\uparrow$) and the collaboration gain spread (CGS) ($\downarrow$) results of our method and other baselines under different dataset partitions. The results are averaged over five independent evaluations.

| Partition | Dataset | FedAvg-FT | | CGSV | | IAFL | | Aequa | |
|---|---|---|---|---|---|---|---|---|---|
| | | MCG | CGS | MCG | CGS | MCG | CGS | MCG | CGS |
| Homogeneous | MNIST | 1.58 | 0.29 | −6.63 | 0.18 | 1.64 | 0.18 | 1.64 | **0.04** |
| | FMNIST | 3.05 | 0.77 | −7.90 | 0.68 | 3.13 | 0.68 | 3.43 | **0.17** |
| | SVHN | 5.08 | 0.81 | −5.26 | 0.39 | 6.57 | 0.39 | 6.68 | **0.08** |
| | CIFAR-10 | 13.62 | 0.65 | −15.08 | 0.48 | 13.62 | 0.59 | 13.31 | **0.09** |
| | CIFAR-100 | 31.54 | 0.64 | −2.02 | 0.43 | 31.54 | 0.56 | 31.14 | **0.14** |
| | SST | 8.77 | 1.48 | 4.74 | 0.81 | 8.96 | 0.81 | 8.11 | **0.22** |
| Heterogeneous: Dirichlet ($\alpha = 0.1$) | MNIST | 29.27 | 14.60 | 43.72 | 12.46 | 16.41 | **11.77** | 47.63 | 12.50 |
| | FMNIST | 15.56 | 14.75 | 31.43 | 9.80 | 8.48 | 10.21 | 40.89 | **9.07** |
| | SVHN | 11.97 | 12.15 | 36.77 | 11.81 | 9.47 | **9.63** | 49.85 | 9.92 |
| | CIFAR-10 | 25.51 | 16.13 | 7.63 | 7.00 | 3.77 | 6.87 | 39.91 | **4.73** |
| | CIFAR-100 | 41.13 | 3.02 | 8.09 | 1.91 | 10.79 | 4.99 | 40.65 | **0.39** |
| | SST | 9.19 | 3.80 | −0.89 | 2.24 | 1.26 | 1.01 | 8.65 | **0.88** |
| Heterogeneous: Dirichlet ($\alpha = 0.5$) | MNIST | 9.48 | 7.37 | 7.27 | 6.00 | 8.93 | **4.59** | 11.56 | 5.23 |
| | FMNIST | 6.81 | 4.60 | 6.24 | 5.92 | 5.86 | 6.05 | 14.34 | **4.41** |
| | SVHN | 12.90 | **5.85** | 14.37 | 7.75 | 12.85 | 6.11 | 21.79 | 6.33 |
| | CIFAR-10 | 28.88 | 9.52 | 6.34 | 6.56 | 13.97 | **3.26** | 28.25 | 5.73 |
| | CIFAR-100 | 37.35 | 1.74 | 4.99 | 1.23 | 27.71 | 7.99 | 37.00 | **0.23** |
| | SST | 9.81 | 2.45 | −0.15 | 2.53 | 3.52 | 1.48 | 9.49 | **1.15** |
| Heterogeneous: Dirichlet ($\alpha = 1.0$) | MNIST | 4.69 | 2.84 | −0.38 | 2.75 | 4.55 | 2.53 | 5.19 | **2.10** |
| | FMNIST | 4.20 | 2.55 | −1.17 | 2.74 | 4.35 | 2.39 | 7.06 | **1.22** |
| | SVHN | 8.10 | 7.11 | 4.08 | 4.53 | 9.72 | 3.38 | 13.36 | **3.13** |
| | CIFAR-10 | 22.69 | 6.54 | −2.38 | 4.75 | 14.94 | 3.50 | 20.39 | **3.50** |
| | CIFAR-100 | 35.46 | 2.00 | 3.37 | 1.22 | 31.08 | 2.94 | 34.60 | **0.22** |
| | SST | 9.48 | 1.97 | 0.77 | 2.12 | 4.74 | 1.73 | 8.50 | **0.88** |
| Heterogeneous: Dirichlet ($\alpha = 2.0$) | MNIST | 2.64 | 1.10 | −3.39 | 1.13 | 2.54 | 1.04 | 2.59 | **0.52** |
| | FMNIST | 2.54 | 2.13 | −3.53 | 1.58 | 3.84 | 1.33 | 4.63 | **0.52** |
| | SVHN | 6.97 | 1.94 | −1.63 | 2.48 | 7.36 | 1.72 | 8.79 | **1.15** |
| | CIFAR-10 | 17.99 | 4.31 | −7.89 | 3.06 | 14.08 | 2.30 | 15.70 | **1.47** |
| | CIFAR-100 | 33.68 | 1.92 | 1.12 | 1.24 | 29.58 | 5.11 | 32.89 | **0.22** |
| | SST | 8.32 | 2.14 | 2.87 | 1.55 | 6.02 | 1.49 | 8.53 | **0.76** |
| Heterogeneous: Dirichlet ($\alpha = 5.0$) | MNIST | 1.84 | 0.40 | −4.23 | 0.37 | 1.78 | 0.32 | 1.65 | **0.09** |
| | FMNIST | 3.05 | 2.58 | −6.74 | 1.62 | 3.74 | 1.64 | 4.45 | **0.67** |
| | SVHN | 5.26 | 4.35 | −1.12 | 1.16 | 6.78 | 0.93 | 7.06 | **0.21** |
| | CIFAR-10 | 15.38 | 3.03 | −11.04 | 2.34 | 9.29 | 7.61 | 13.04 | **0.78** |
| | CIFAR-100 | 32.47 | 1.04 | 0.58 | 0.66 | 31.92 | 1.33 | 32.01 | **0.17** |
| | SST | 8.73 | 1.17 | 3.66 | 1.02 | 7.36 | 1.20 | 8.28 | **0.27** |
| Quantity Skew: Imbalanced $(0.15, 6)$ | MNIST | 2.83 | 2.50 | −2.23 | 2.09 | 2.33 | 1.73 | 2.43 | **1.56** |
| | FMNIST | 3.96 | 2.85 | −4.98 | 2.17 | 3.59 | 1.42 | 3.72 | **0.49** |
| | SVHN | 7.91 | 5.00 | −3.56 | 4.47 | 6.25 | **1.73** | 7.98 | 2.84 |
| | CIFAR-10 | 19.88 | 14.02 | −5.85 | 8.96 | 13.67 | **7.01** | 17.62 | 10.31 |
| | CIFAR-100 | 34.13 | 13.63 | 2.87 | 7.93 | 21.23 | **1.26** | 29.56 | 5.07 |
| | SST | 9.16 | 2.46 | 4.78 | 2.40 | 6.85 | 1.75 | 8.62 | **1.21** |
| Quantity Skew: Imbalanced $(0.4, 2)$ | MNIST | 4.60 | 2.38 | −1.13 | 2.33 | 2.58 | 1.58 | 3.44 | **1.58** |
| | FMNIST | 5.71 | 3.00 | −5.93 | 2.72 | 3.28 | 1.73 | 3.76 | **1.25** |
| | SVHN | 11.80 | 5.04 | 1.30 | 4.77 | 5.94 | **2.21** | 10.08 | 3.49 |
| | CIFAR-10 | 29.65 | 14.70 | −2.20 | 10.60 | 13.88 | **8.03** | 24.88 | 11.76 |
| | CIFAR-100 | 41.66 | 19.02 | 6.92 | 11.84 | 10.63 | **5.59** | 30.96 | 12.42 |
| | SST | 11.08 | 3.56 | 3.63 | 4.16 | 3.85 | **1.69** | 9.64 | 2.39 |
| Label Skew: #OC={3, 30} | MNIST | 18.99 | 10.86 | 50.04 | 0.71 | 17.88 | 11.98 | 64.34 | **0.06** |
| | FMNIST | 11.61 | 9.64 | 34.63 | 2.35 | 13.22 | 9.58 | 51.80 | **0.20** |
| | SVHN | 1.13 | 1.07 | 38.59 | 1.13 | 9.21 | 8.47 | 51.45 | **0.09** |
| | CIFAR-10 | 24.11 | 6.38 | 5.79 | 4.86 | 5.57 | 11.68 | 39.42 | **0.18** |
| | CIFAR-100 | 40.28 | 1.40 | 5.25 | 0.84 | 23.16 | 9.93 | 40.75 | **0.16** |
| | SST | 8.86 | 1.70 | −0.89 | 1.97 | 5.55 | 1.92 | 7.60 | **0.65** |
| Number of times that performs the best | | 1/54 | | 0/54 | | 11/54 | | **42/54** | |

Table 8: The average collaboration gain (MCG) (↑) and the collaboration gain spread (CGS) (↓) results of our method and other baselines under different dataset partitions. The results are averaged over five independent evaluations. Continuation of Table 7 with different minimum width ($p_{\min} = 0.1$).

| PARTITION | DATASET | FEDAVG-FT | | CGSV | | IAFL | | AEQUA | |
|---|---|---|---|---|---|---|---|---|---|
| | | MCG | CGS | MCG | CGS | MCG | CGS | MCG | CGS |
| QUANTITY SKEW: IMBALANCED $(0.15, 6)$ | MNIST | 2.83 | 2.50 | −2.23 | 2.09 | 2.33 | 1.73 | 1.70 | **1.02** |
| | FMNIST | 3.96 | 2.85 | −4.98 | 2.17 | 3.59 | 1.42 | 2.82 | **0.45** |
| | SVHN | 7.91 | 5.00 | −3.56 | 4.47 | 6.25 | 1.73 | 5.38 | **0.75** |
| | CIFAR-10 | 19.88 | 14.02 | −5.85 | 8.96 | 13.67 | 7.01 | 13.46 | **3.04** |
| | CIFAR-100 | 34.13 | 13.63 | 2.87 | 7.93 | 21.23 | 1.26 | 24.72 | **1.16** |
| | SST | 9.16 | 2.46 | 4.78 | 2.40 | 6.85 | 1.75 | 7.71 | **0.58** |
| QUANTITY SKEW: IMBALANCED $(0.4, 2)$ | MNIST | 4.60 | 2.38 | −1.13 | 2.33 | 2.58 | 1.58 | 2.59 | **1.28** |
| | FMNIST | 5.71 | 3.00 | −5.93 | 2.72 | 3.28 | 1.73 | 2.69 | **1.09** |
| | SVHN | 11.80 | 5.04 | 1.30 | 4.77 | 5.94 | 2.21 | 5.89 | **1.71** |
| | CIFAR-10 | 29.65 | 14.70 | −2.20 | 10.60 | 13.88 | 8.03 | 14.20 | **6.62** |
| | CIFAR-100 | 41.66 | 19.02 | 6.92 | 11.84 | 10.63 | **5.59** | 21.08 | 7.64 |
| | SST | 11.08 | 3.56 | 3.63 | 4.16 | 3.85 | 1.69 | 8.80 | **1.66** |
| NUMBER OF TIMES THAT PERFORMS THE BEST | | 0/12 | | 0/12 | | 1/12 | | **11/12** | |

Table 9: Incentivization performance of Aequa on CIFAR-10, CIFAR-100, and MNIST datasets under various partitioning strategies, evaluated using the Pearson correlation coefficient between (1) assigned model widths and contributions and (2) final model accuracies and contributions. Results are averaged over five independent runs.

| DATASET | PARTITION | $\rho(\boldsymbol{c}, \boldsymbol{p})$ (CONTRIBUTION, WIDTH) | $\rho(\boldsymbol{c}, \boldsymbol{a})$ (CONTRIBUTION, ACC.) |
|---|---|---|---|
| CIFAR-10 | HOMOGENEOUS | $1.00 \pm 0.00$ | $0.94 \pm 0.01$ |
| | DIRICHLET $(\alpha = 0.1)$ | $1.00 \pm 0.00$ | $0.91 \pm 0.04$ |
| | DIRICHLET $(\alpha = 0.5)$ | $1.00 \pm 0.00$ | $0.95 \pm 0.02$ |
| | DIRICHLET $(\alpha = 1.0)$ | $1.00 \pm 0.00$ | $0.95 \pm 0.02$ |
| | DIRICHLET $(\alpha = 2.0)$ | $1.00 \pm 0.00$ | $0.96 \pm 0.02$ |
| | DIRICHLET $(\alpha = 5.0)$ | $1.00 \pm 0.00$ | $0.92 \pm 0.04$ |
| | QUANTITY SKEW $(0.15, 6)$ | $1.00 \pm 0.00$ | $1.00 \pm 0.00$ |
| | QUANTITY SKEW $(0.4, 2)$ | $1.00 \pm 0.00$ | $0.99 \pm 0.01$ |
| | LABEL SKEW $(\#C = \{3, 30\})$ | $1.00 \pm 0.00$ | $0.96 \pm 0.02$ |
| CIFAR-100 | HOMOGENEOUS | $1.00 \pm 0.00$ | $0.95 \pm 0.01$ |
| | DIRICHLET $(\alpha = 0.1)$ | $1.00 \pm 0.00$ | $0.96 \pm 0.02$ |
| | DIRICHLET $(\alpha = 0.5)$ | $1.00 \pm 0.00$ | $0.92 \pm 0.02$ |
| | DIRICHLET $(\alpha = 1.0)$ | $1.00 \pm 0.00$ | $0.94 \pm 0.03$ |
| | DIRICHLET $(\alpha = 2.0)$ | $1.00 \pm 0.00$ | $0.95 \pm 0.03$ |
| | DIRICHLET $(\alpha = 5.0)$ | $1.00 \pm 0.00$ | $0.96 \pm 0.02$ |
| | QUANTITY SKEW $(0.15, 6)$ | $1.00 \pm 0.00$ | $1.00 \pm 0.00$ |
| | QUANTITY SKEW $(0.4, 2)$ | $1.00 \pm 0.00$ | $1.00 \pm 0.00$ |
| | LABEL SKEW $(\#C = \{3, 30\})$ | $1.00 \pm 0.00$ | $0.95 \pm 0.02$ |
| MNIST | HOMOGENEOUS | $1.00 \pm 0.00$ | $0.91 \pm 0.04$ |
| | DIRICHLET $(\alpha = 0.1)$ | $1.00 \pm 0.00$ | $0.90 \pm 0.03$ |
| | DIRICHLET $(\alpha = 0.5)$ | $1.00 \pm 0.00$ | $0.92 \pm 0.04$ |
| | DIRICHLET $(\alpha = 1.0)$ | $1.00 \pm 0.00$ | $0.92 \pm 0.03$ |
| | DIRICHLET $(\alpha = 2.0)$ | $1.00 \pm 0.00$ | $0.94 \pm 0.04$ |
| | DIRICHLET $(\alpha = 5.0)$ | $1.00 \pm 0.00$ | $0.88 \pm 0.03$ |
| | QUANTITY SKEW $(0.15, 6)$ | $1.00 \pm 0.00$ | $0.95 \pm 0.04$ |
| | QUANTITY SKEW $(0.4, 2)$ | $1.00 \pm 0.00$ | $0.94 \pm 0.04$ |
| | LABEL SKEW $(\#C = \{3, 30\})$ | $1.00 \pm 0.00$ | $0.90 \pm 0.04$ |

