# OpenReview forum: "Aequa: Fair Model Rewards in Collaborative Learning via Slimmable Networks"
_ICML.cc/2025/Conference — ICML 2025 poster_

### Official Review · Reviewer_gzyb · 2025-03-13

**Overall Recommendation:** 3

**Summary:**

Achieving collaborative fairness in federated learning involves contribution assessment and reward allocation mechanisms. This paper proposes a new reward mechanism that leverages slimmable neural networks (a client with lower contribution would get a neural network of smaller width and lower accuracy). Aequa can decide reward values for post-training distribution of model rewards and also be adapted as part of training time model rewards.

## update after rebuttal
The authors have addressed my concerns and I raised my score during the rebuttal process.

**Claims And Evidence:**

Mostly, the objectives of an ideal allocation algorithm (in Sec 4.2) should be better justified. For example, reducing the variability may actually help free riders. Equation 3 is only one way to achieve these objectives. There are others concepts that consider both fairness and efficiency such as Nash social welfare (which consider the product of utilities) and lp-norms.

**Essential References Not Discussed:**

Under reward mechanisms, Lin et al, 2023 should also be discussed and compared against.

**Experimental Designs Or Analyses:**

I checked the soundness of any experimental designs or analyses.
* It is better to only change one factor at a time, e.g., replace zeroing out gradients with slimmable neural networks and see whether the trend in accuracy is better. Fix the reward mechanism that decide reward values (e.g., proportional to shapley value) and show that the resulting model reward accuracy is more correlated to the Shapley values.

**Methods And Evaluation Criteria:**

Mostly, I am unsure about post-training model rewards as each client would have got the best possible full model during training based on Algorithm 1 (without TEEs).

**Other Comments Or Suggestions:**

* The Impact Statement is missing
* Move algorithm 2 to the main paper. Specify the reward mechanism
* Instead of claiming aqua to be "agnostic to any contribution measure",  it might be more accurate to say that it is a reward mechanism (and complements any contribution assessment step).

**Other Strengths And Weaknesses:**

**Strengths**
1. The paper introduces a novel reward allocation mechanism by leveraging slimmable neural networks.
2. The paper is generally well-written and easy to understand.

**Weaknesses**
1. I assume that the fair allocation algorithm is directly applied after algorithm 1. **However, if algorithm 1 is used, each client would already get the full model parameters.** As the authors have pointed out in the introduction (line 62), this aids free-riding and does not ensure collaborative fairness subsequently.
    * This limitation is partially addressed by TEEs
    * In Sec 4.3, an alternative reward mechanism (Equation 5) is proposed. **Is this used in place of Equation 3 when training-time rewards are desired?**
2. The paper is not self-contained and important details (e.g., algorithm in Sec 4.3, proofs in Sec 5, experiment results) are left to the appendix). It would be better to incorporate them in the main paper, for example, give the intuition behind some more complex proofs (e.g, Theorem 2) in the main paper.

**Questions For Authors:**

1. In Sec 6.2 and 6.3, is Aequa referring to combining Algorithm 1 and Sec 4.2? What is Aequa reward mechanism in Sec 6.3? How do you decide the model width for each client?
2. Address weakness 1 and in particular, the bolded statements.
4. Can you provide some intuition and description about the setting in Theorem 2? Is it still the post training reward setting? Why are multiple iterations considered?

**Relation To Broader Scientific Literature:**

The paper proposes an alternative reward mechanism, slimmable neural networks, for federated learning. Existing works have considered sparsifying model parameters, only sharing updates from a few clients, and controlling the frequency of the client receiving updates.

**Theoretical Claims:**

I only checked the correctness of Sec 5.2.

---

> ### Author Rebuttal · Authors · 2025-04-01
>
> > Experimental Designs
>
> **E1:** We appreciate the reviewer's effort in scrutinizing the experimental design. However, we would like to clarify an **important point**: there is no alternative definition of collaborative fairness in our work or in the literature - the goal is consistent across methods. Our objective function in Sec. 4.2 aims to reduce the variance in $u(\mathbf{a})$, which promotes equitable benefit distribution and is a desirable property. Even without using CGS, Aequa achieves high Pearson correlation scores, reinforcing its fairness. Our formulation is principled and aligned with Tastan et al., 2025.
>
> **E2:** Thank you for this comment. Our design targets a different setting than suggested. When using contribution assessment methods like CGSV - which uses the *zeroing out gradients* strategy - we already test the scenario the reviewer proposes. As shown in Figs. 3 & 4, Aequa (CGSV) ranks second-best in correlation and CGS, validating our approach.
>
> ---
> > References
>
> We will include a discussion of Lin et al., 2023 in the final manuscript. However, that work focuses on balancing performance and collaborative fairness, unlike ours, which aims to maximize fairness directly. Their results do not match the high fairness levels achieved by Aequa (please refer to the paper), but we acknowledge its relevance and will cite it in Sec. 2.
>
> ---
> > Weaknesses
>
> We appreciate the reviewer's feedback but would like to clarify a key aspect that may have been **overlooked**. Our method explicitly assumes the use of TEEs for local client training, preventing clients from accessing full model parameters and thereby preventing free-riding concerns.
>
> We kindly request the reviewer to revisit the paper for further clarification and consider adjusting their assessment accordingly.
>
> **W1.3**: The reviewer’s understanding is correct. Eq. 5 is used in training-time reward settings for simplicity.
>
> **W2.1**: Our method can incorporate contribution-based strategies. Free riders can be assigned the minimum-width model (even set to zero). The utility definition (allocation minus contribution) can also be formulated as a ratio without affecting Lemmas 3 & 4 (it is straightforward to prove).
>
> As for the case where a client's contributions equals $u$, it represents a quantity skew - a common FL challenge. Our experiments include such scenarios. Increasing the minimum width parameter adjusts allocations for lower-contributing clients.
>
> Aequa aligns with Lyu et al., 2020's fairness principle: allocated reward proportional to contributions.
>
> **W2.2**: While we don't follow prior procedures exactly, our minimum width parameter functions similarly to the existing tradeoff parameters the reviewer mentioned.
>
> When comparing Aequa to other works, we used their best-performing parameters, as detailed in Appendix B. For CGSV, we used the best setting ($\beta=1.0$), as justified in the original paper.
> Moreover, Aequa's best-performing clients end up with a comparable or better model than the model obtained by simply running  FedAvg, making an explicit tradeoff parameter unnecessary - a strength of our approach.
>
>
> **W2.3**: We would like to highlight a fundamental misunderstanding in the reviewer's assessment.
>
> First, predictive performance is unrelated to Eq. 3, as this equation is applied post-training and does not influence the model's learning process. Any suggestion otherwise misinterprets our approach.
>
> Second, there is no "alternative" definition of collaborative fairness. The concept remains consistent across works; the only difference lies in how it is operationalized. Our objective aligns with Tastan et al., 2025, as previously stated.
>
> Finally, we urge the reviewer to carefully examine also the Pearson correlation results, where Aequa consistently outperforms all baselines. Our method is both theoretically justified and empirically validated, reinforcing its effectiveness in achieving collaborative fairness.
>
> **W3:** Due to page limitations, we included only the most essential components of our work in the main paper while placing proofs and extended experiments in the appendix - a common practice in the literature. However, we acknowledge the reviewer's suggestion and will incorporate additional details in the main text where feasible. We will use an extra page for camera-ready for this.
>
> > Comments or Suggestions
>
> We appreciate the reviewer’s suggestions and will incorporate them.
>
> > Questions
>
> **Q1:** Yes. Based on the reward mechanism in Section 4.2.
>
> **Q4:** Yes, Theorem 2 applies to the post-training reward setting, where the contributions of each participant and the allocation vector are already known. The multiple iterations are used to determine $\mathbf{a}^{\star}$ that minimizes the objective defined in Eq. 3, i.e., we run an iterative optimization algorithm to determine optimal widths for each client without updating the model and only taking client contributions as an input.

---

> > ### Comment · Reviewer_gzyb · 2025-04-02
> >
> > > We appreciate the reviewer's feedback but would like to clarify a key aspect that may have been overlooked. Our method explicitly assumes the use of TEEs for local client training, preventing clients from accessing full model parameters and thereby preventing free-riding concerns.
> >
> > Thank you for the clarification! This improves the opinion of my work but should also be included beyond Section 1.
> >
> > > First, predictive performance is unrelated to Eq. 3, as this equation is applied post-training and does not influence the model's learning process. Any suggestion otherwise misinterprets our approach.
> >
> > My interpretation is as follows: Minimising Eq. 3 would mean that your allocation vector would reward free riders with more valuable models (post-training). Can you clarify this further?
> >
> > > Finally, we urge the reviewer to carefully examine also the Pearson correlation results, where Aequa consistently outperforms all baselines.
> >
> > Thank you for the clarification! In data valuation, fairness is usually based on proportionality to individual/Shapley values. Initially, I found it strange that the Pearson correlation results is a measure of fairness as it suggests that rewarding clients by their individual standalone accuracy is optimal.
> > Upon re-checking the related work (Xu 2021, Wu 2024), I noticed that in FL, fairness can be approximately assessed by the correlation with standalone accuracy. Indeed, Aequa outperforms existing methods.
> >
> > ---
> > The additional rebuttal comments has addressed my concerns. I have removed the weakness and raised the score from 2 to 3.

---

> > > ### Author Response · Authors · 2025-04-03
> > >
> > > Thank you for the follow-up. We respectfully disagree that minimizing Equation 3 rewards free riders with more valuable models. Maybe the confusion arises due to misinterpretation of the utility $u$ as the accuracy of the allocated models. We would like to emphasize that utility $u$ is defined as **collaboration gain** (difference between the allocation $a$ and the contribution $c$) in Equation 2. While the numerator in Equation 3 (expected utility/collaboration gain) is maximized when all the clients receive more valuable models (irrespective of their contribution), such an allocation would greatly increase the denominator (variance of the collaboration gains) when there are free-riders. Consequently, such an allocation will not minimize Equation 3. Rather, Equation 3 is minimized when the variance of the collaboration gains is small (collaboration gains are uniform), i.e., free riders receive models with lesser width (and hence lower accuracy) **and** high contributors receive models with higher width (and hence better accuracy), while at the same time ensuring that expected collaboration gain stays positive, i.e., all clients receive models whose accuracy is better than their respective standalone accuracies.
> > >
> > > In particular, if a client has a near-zero contribution (free rider), the optimal solution to Eq. 3 assigns that client a low utility model, i.e., a minimum-width model with low accuracy. On the other hand, a client with the highest contribution receives the full-width model with high accuracy. In both these scenarios, the collaboration gain is expected to be positive and the variance of the collaboration gains would be very low, thereby minimizing the objective in Eq. 3.
> > >
> > > ---
> > >
> > > If the concern of the reviewer pertains to the case where the accuracy of the minimum-width model is still too high (thereby rewarding free riders), we recommend lowering the minimum width parameter. This ensures that free riders do not benefit disproportionately, maintaining strict contribution-based fairness. If this is indeed the question, we would like to direct the reviewer to our quantity skew experiments presented in the main paper. This setting includes clear free-rider scenarios. For instance:
> > >
> > > - In quantity skew (0.15, 6), four participants each hold only $2.5$% of the data (free riders), while six participants each hold $15$%.
> > > - In quantity skew (0.4, 2), eight participants hold $2.5$% each, while two high-contributing clients hold $40$% each.
> > >
> > > These cases are detailed in the experiments, specifically in Tables 1, 2, 6, and 7. In Table 6, we use a minimum width of **0.25**. While Aequa performs well in general, it underperforms compared to IAFL in these extremely skewed partitions in terms of collaboration gain metrics (CGS, but perfect in performance and Pearson correlation).
> > >
> > > To address this, in Table 7, we reduce the minimum width to **0.1**. This adjustment significantly decreases the gains of free riders (decreases the performance of the minimum-width model). As a result, Aequa outperforms all baselines across CGS (variance), Pearson correlation, and predictive performance, achieving near-perfect fairness and accuracy.
> > >
> > > These results demonstrate that even in extreme skew scenarios (scenarios with free riders), the solution to Eq. 3, in conjunction with an appropriate minimum width setting, prevents rewarding free riders and remains faithful to collaborative fairness.

---

### Official Review · Reviewer_wyh2 · 2025-03-14

**Overall Recommendation:** 3

**Summary:**

The paper introduces a framework (Aequa) for fair model rewards in collaborative learning (CL) by leveraging slimmable neural networks. The core idea is to proportionally allocate model capacity to participants based on their contributions, rather than distributing identical models to all. The method ensures that higher contributors receive better-performing models (e.g., a neural network with a bigger width), while lower contributors get degraded versions.

**Claims And Evidence:**

supported

**Essential References Not Discussed:**

References ok.

**Experimental Designs Or Analyses:**

valid

**Methods And Evaluation Criteria:**

probably sound

**Other Comments Or Suggestions:**

NA

**Other Strengths And Weaknesses:**

Strengths

1. Integrating slimmable networks (a single neural network that can operate at multiple widths) with federated learning for fairness is unique. Unlike previous methods that use heuristic reward distributions, Aequa directly controls model width, ensuring a structured degradation in performance.

2. Theoretical Convergence Analysis
The framework provides a comprehensive convergence analysis to ensure the optimization of the training-time reward allocation algorithm remains stable and optimal. This is a notable improvement over methods like CGSV, which lack formal convergence proofs, and IAFL, which introduces trade-offs between fairness and performance.



Weaknesses
1.   Assumption of Continuous Model Performance: The theoretical analysis assumes that model performance is continuous within the interval [ℓ, u], allowing for perfect Pearson correlation coefficients. However, in practice, model widths are discrete due to hardware and software limitations, and performance may not scale smoothly with width reduction. This discreteness could lead to suboptimal allocations, where small changes in width result in disproportionate performance drops, affecting fairness.

2. There are some missing refs in related fields such as [1][2].

3. Limited Scalability to Large Federated Learning Setups
The approach has only been tested on relatively small datasets (CIFAR-10, MNIST, etc.), which do not reflect the scale of real-world FL scenarios (e.g., federated medical imaging, large-scale NLP). Lack of experiments on larger FL benchmarks: The method should be evaluated on federated benchmarks like FEMNIST or OpenImage-FL to test scalability.


[1] Jiang, M., Roth, H. R., Li, W., Yang, D., Zhao, C., Nath, V., ... & Xu, Z. (2023). Fair federated medical image segmentation via client contribution estimation. In Proceedings of the IEEE/CVF Conference on Computer Vision and Pattern Recognition (pp. 16302-16311).
[2] Li, T., Hu, S., Beirami, A., & Smith, V. (2021, July). Ditto: Fair and robust federated learning through personalization. In International conference on machine learning (pp. 6357-6368). PMLR.

**Questions For Authors:**

NA

**Relation To Broader Scientific Literature:**

Fair reward allocation is important in federated learning.

**Theoretical Claims:**

probably sound

---

> ### Author Rebuttal · Authors · 2025-04-01
>
> We thank the reviewer for taking the time to review our paper, for highlighting its strengths, and for their valuable feedback.
>
> > W1. Continuous model performance assumption
>
> We acknowledge that the assumption of continuous model performance across the interval $[\ell, u]$ may not always hold perfectly in practice. This assumption was primarily introduced for theoretical rigor. Nevertheless, as indicated and supported by other reviewers, our experimental results consistently achieve fairness scores close to the perfect score (mostly exceeding 0.95). Based on these comprehensive experiments, we confidently assert that the practical discreteness of model widths does not significantly impact performance or fairness, as our fairness scores consistently remain very high, often approaching or equaling 1.0.
>
> > W2. Missing references
>
> Please note that reference [1] is already cited in our paper (lines 40, 59, 90).
> Regarding [2], while we acknowledge its relevance in the broader context of fairness in federated learning, its focus differs from the core objective of our work. [2] specifically targets performance fairness, aiming to personalize models for individual clients to ensure each achieves reasonable predictive performance. In contrast, our work addresses collaborative fairness, which is concerned with the fair distribution of model rewards in proportion to each participant's contribution.
>
> Given this, [2] is not central to our framework. Nonetheless, to ensure completeness in the extended related work discussion, we will include it in Section 2 (Fairness in FL) of the final version of the paper.
>
>
> > W3. Scalability experiments
>
> | Algorithm | Acc. | $\rho$ | MCG $\pm$ CGS |
> | --- | --- | --- | --- |
> | FedAvg-FT | 71.48 | 0.2904 | 47.02 $\pm$ 5.07 |
> | Aequa (ours) | **73.10** | **0.9888** | 52.19 $\pm$ **2.57** |
> [FEMNIST experiment]
>
> In response to the reviewer's suggestion, we conducted additional scalability experiments on the FEMNIST dataset to evaluate the effectiveness of our proposed method under challenging federated learning conditions. FEMNIST is a large-scale benchmark dataset characterized by its natural non-IID distribution and extensive client base, making it well-suited for evaluating scalability.
>
> For this experiment, we employed a custom CNN architecture composed of two convolutional layers followed by a fully connected layer. The experiment involved a total of $\textbf{3597}$ **clients** with partial participation, randomly sampling 10 clients per communication round. We set the number of local epochs to 1, a batch size of 16, and executed training for 500 communication rounds. The training was performed using an SGD optimizer with an initial learning rate of 0.1, employing the same settings detailed in the main paper.
>
> The results in the table demonstrate that our method consistently outperforms the FedAvg algorithm across all evaluation metrics. This indicates that Aequa effectively addresses the scalability and heterogeneity challenges inherent in large-scale federated learning scenarios.
>
> We also include the model width vs. accuracy plot in this link: https://ibb.co/C3jpd5Nm

---

### Official Review · Reviewer_iRNs · 2025-03-19

**Overall Recommendation:** 3

**Summary:**

The manuscript studies the question of assigning rewards to participants with different models whose performance faithfully reflects their heterogeneous contribution, and extends/repurposes the concept of the slimmable network for fairness in federated learning, so as to make sure that model rewards are proportional to client contributions, achieving both high performance and collaborative fairness simultaneously.

**Claims And Evidence:**

Yes. The manuscript includes theoretical convergence results and fairness analysis, as well as extensive numerical results to support the advances of the proposed method.

**Essential References Not Discussed:**

NA

**Experimental Designs Or Analyses:**

Yes. The numerical experiments look strong.

**Methods And Evaluation Criteria:**

Looks reasonable:
* it considers MNIST, Fashion-MNIST, SVHN, CIFAR-10 & 100;
* it uses homogeneous, heterogeneous, and quantity skew scenarios;
* it also considers several baselines, including CGSV, IAFL, SA.

**Other Comments Or Suggestions:**

Lemma 4 is very hard to understand, as it does not explain the meaning of $\mathbf{c}$ and $\rho$.

## update after rebuttal
The reviewer thanks the authors for providing feedback and will maintain the original score.

**Other Strengths And Weaknesses:**

NA

**Questions For Authors:**

NA

**Relation To Broader Scientific Literature:**

NA

**Theoretical Claims:**

* The proof of the convergence analysis looks standard, and it is unclear how much the subnetwork will affect the optimization.
* The writing quality of sec 5.2 should be improved.

---

> ### Author Rebuttal · Authors · 2025-04-01
>
> We thank the reviewer for taking the time to review our work, for highlighting its advantages and for their valuable positive feedback.
>
> > The writing quality of Section 5.2 should be improved.
>
> We appreciate the reviewer highlighting the clarity issues in Section 5.2. We sincerely apologize for any confusion caused and have carefully revised this section to improve readability and coherence. Additionally, we have ensured that all important points previously omitted are clearly incorporated into the final manuscript.
>
> > How much the subnetwork will affect the optimization?
>
> As we show in Lemmas 1 and 2, subnetworks preserve both smoothness and convexity of the objective. Thus, the effect on the optimization is minimal, and our new formulation admits standard analysis as demonstrated in our manuscript, which is the strong point. Therefore, the main effect that we need to investigate is the final quality of models given subnetwork formulation, which consistently maintains strong performance as validated by our experimental results.
>
> > Lemma 4
>
> We apologize for the oversight in defining the terms used in Lemma 4. We clarify that $\mathbf{c}$ represents the vector containing individual client contributions, as elaborated in Section 4.2. Additionally, $\rho$ denotes the Pearson correlation (line 281). We appreciate the reviewer's comment and will revise Lemma 4 and the explanation to incorporate these clarifications in the final manuscript.

---

### Official Review · Reviewer_yhe8 · 2025-03-22

**Overall Recommendation:** 4

**Summary:**

This paper introduces Aequa, a framework to ensure collaborative fairness in federated learning using slimmable neural networks. It trains a single global model whose sub-networks of varying widths serve as rewards aligned with participant contributions. Experiments on six benchmark datasets show Aequa achieves near-perfect correlation between contributions and model performance without significant loss in overall accuracy.

**Claims And Evidence:**

The authors claim Aequa ensures proportional rewards, maintains strong performance, and converges theoretically. Correlation results (often above 0.95) back up the fairness claim, and tables show minimal accuracy drop compared to FedAvg. The theoretical analysis relies on standard convex FL assumptions; while the reviewer did not verify every proof step, the logic appears sound.

**Essential References Not Discussed:**

No major omissions are evident; the authors cite relevant literature on fair FL and slimmable architectures.

**Experimental Designs Or Analyses:**

Experiments include varied data partitions (Dirichlet, label/quantity skew) and compare multiple metrics. Aequa consistently achieves the highest fairness correlation while matching or improving baseline accuracies. A few corner cases require tuning the minimum width parameter.

**Methods And Evaluation Criteria:**

The methods and experimental setup appear well-chosen for the problem. The core method – using a slimmable network to enable differentiated model widths – is appropriate because it allows implementing fairness in a single training run (as opposed to training separate models per client). This design is elegant and efficient, ensuring that smaller “reward” models are true sub-networks of the larger model and thus require no additional training ￼. The paper’s federated optimization procedure (Algorithm 1) is an adaptation of FedAvg to train all widths of the slimmable model in a coordinated way; this is a sensible approach to ensure that the global model performs well across the full width range. The fair allocation algorithm (post-training) is described formally as an optimization problem with clear objectives (non-negativity of gains, proportionality, etc.), and the chosen solution method (simulated annealing-based heuristic) is reasonable for finding an approximate optimal width allocation. The evaluation criteria align with the claims: the authors evaluate global model accuracy to ensure overall performance isn’t degraded, and they evaluate fairness through the correlation between contributions and rewards (as well as a metric called collaboration gain spread). These metrics directly measure the goals of collaborative fairness and are standard in this line of work ￼. The use of six diverse datasets (MNIST, FMNIST, SVHN, CIFAR-10, CIFAR-100, SST) and multiple data partition strategies (homogeneous, Dirichlet heterogeneous, quantity skew, label skew) is commendable ￼ ￼. This covers both IID and non-IID scenarios, which is critical for federated learning experiments. The chosen baselines are appropriate: FedAvg with fine-tuning represents a naive approach to personalization, CGSV and IAFL represent state-of-the-art fairness/incentive methods ￼, and evaluating Standalone accuracy provides a reference point for each client’s individual performance. By including these baselines, the authors ensure a fair and informative comparison. The experiments were run across all methods on identical settings, and the paper mentions using balanced accuracy for class-imbalanced data which is a proper choice ￼. In terms of methodology, everything from the training procedure to the selection of metrics seems well-justified and in line with common practice, indicating the experimental design is suitable for validating the paper’s contributions.

**Other Comments Or Suggestions:**

Strengths include a novel architectural approach, solid empirical validation, and flexible application to different contribution measures. Weaknesses are its reliance on TEEs for security and the need to choose parameters (e.g., minimum model width) carefully. Further exploration of real-world overhead and security challenges would be beneficial.

**Other Strengths And Weaknesses:**

Strengths include a novel architectural approach, solid empirical validation, and flexible application to different contribution measures. Weaknesses are its reliance on TEEs for security and the need to choose parameters (e.g., minimum model width) carefully. Further exploration of real-world overhead and security challenges would be beneficial.

**Questions For Authors:**

(1) How does Aequa handle noisy or imperfect contribution assessments?

(2) Could Aequa mitigate free-riding if trusted hardware is unavailable, or is TEE integral?

(3) Does simultaneously training many width configurations introduce significant overhead for very large models or client counts?

**Relation To Broader Scientific Literature:**

The paper builds on existing FL incentive work (e.g., CGSV, IAFL) but uniquely applies slimmable networks to allocate capacity. It aligns with “collaborative fairness” studies, providing a more rigorous and flexible mechanism than many prior heuristic methods.

**Theoretical Claims:**

The paper makes two primary theoretical claims: a performance guarantee for the federated training process with slimmable networks, and the convergence of the fair allocation algorithm. Both are presented with formal statements (Theorem 1 and Theorem 2) and proved under certain assumptions. The correctness of these claims appears plausible – the authors assume standard conditions (convex loss, L-smoothness, bounded variance in gradients, etc., as referenced in the paper) and then build on known federated optimization analyses ￼. Theorem 1 provides a convergence rate or error bound for Aequa’s training algorithm, ensuring that training on slimmable networks still optimizes the global objective within the same order of convergence as regular federated learning (given a suitable learning rate) ￼. Theorem 2 addresses the allocation algorithm, stating that the iterative simulated annealing approach will converge asymptotically to the optimal allocation (with probability 1) ￼. The proofs for these are relegated to the appendix, and the authors reference lemmas and prior work (e.g. citing known results from convex optimization literature) to support their derivations. From a clarity standpoint, the theoretical section in the main text is relatively concise – it outlines key lemmas and theorems, deferring detailed technical proofs to Appendix A. This makes it a bit challenging to verify every step without deep diving into the supplementary material, but it keeps the main content accessible. The reviewer’s confidence in the theoretical claims is moderate rather than absolute: the reasoning is sound on a high level and no obvious errors were found, but a full verification would require checking each appendix lemma and assumption carefully. The authors do clearly state assumptions (e.g., convexity and bounded client-drift via bounded dissimilarity in data, as noted in the appendix references) and the statements of theorems seem consistent with those assumptions. The convergence claim for the allocation algorithm is particularly interesting since it uses a heuristic method (simulated annealing); proving convergence in that context is non-trivial, but the authors appear to have done so under specific conditions (assuming a certain form of cost function and proper cooling schedule, presumably detailed in Appendix A.6). In summary, the theoretical claims are well-motivated and likely correct, though their practical applicability is bound by the validity of the assumptions (which is typical for FL theory). The clarity is sufficient, but readers who want full detail will need to consult the supplementary proofs, which the reviewer trusts with some caution.

---

> ### Author Rebuttal · Authors · 2025-04-01
>
> We sincerely thank the reviewer for their valuable positive feedback.
>
> > Reliance on TEEs for security
>
> We acknowledge the reviewer's concern regarding reliance on TEEs. This limitation is already discussed in the paper, in Section 7. While TEE is a necessity for Aequa, Aequa can also operate in a setting without trusted hardware by incorporating contribution assessment methods (e.g., CGSV). In this case, model allocations are directly tied to assessed contributions, ensuring fairness without requiring hardware-based isolation.
>
> Thus, while TEEs provide one implementation path and theoretically fair algorithm, Aequa remains effective and secure even in their absence.
>
> > Need to choose parameters (e.g. minimum model width) carefully.
>
> We acknowledge the reviewer's concern regarding the careful selection of parameters. However, our proposed method notably differs from existing approaches regarding hyperparameter sensitivity. Unlike existing methods, which typically rely on one or two hyperparameters that must be carefully tuned to balance fairness and utility, our method utilizes only a single parameter - the minimum model width.
>
> Importantly, we do not treat this parameter as a conventional hyperparameter requiring extensive tuning because it has a clear, interpretable meaning: it represents the minimum model width corresponding to the lowest performance model. Furthermore, setting this parameter to its minimal feasible value does not have an effect on both predictive performance and fairness metrics, thereby significantly reducing sensitivity and simplifying practical deployment.
>
> > Further exploration of real-world overhead and security challenges
>
> Please take a look at the response to [Reviewer wyh2, W3] to see the results on the FEMNIST dataset that effectively mimics the real-world overhead and how our method performs on all evaluation metrics. We supplement the training time per communication round in seconds from the provided table.
>
> | Algorithm | time (s) |
> | ---- | ----- |
> | FedAvg | 42 $\pm$ 2.9 s |
> | Aequa | 55 $\pm$ 3.1 s |
> |
>
> From this table, FedAvg completes a round in 42 seconds, whereas Aequa takes 55 seconds - an increase of 1.309 times or 30.9% over FedAvg. This overhead remains modest, not significant, and is well justified by Aequa's superior performance and fairness. As per the communication overhead, there is no overhead as we communicate the same-sized models.
>
> Regarding security challenges, the main consideration is the requirement of TEEs on client devices, as described in Section 1. However, the training-time extended version of Aequa, described in Section 4.3, does not depend on TEEs. In such a setting, broadcasted model widths from the server are based on contribution scores, mitigating security concerns related to model access.
>
>
> > Questions
>
> **Q1:** In this work, we specifically focus on the reward allocation mechanism, operating under the assumption that the contribution assessments provided are reliable. Addressing improvements in contribution assessment methods falls beyond the current scope of our study.
>
> **Q2:** Aequa's design inherently supports mitigating free-riding even in the absence of trusted hardware (TEE). By integrating robust contribution assessment methods (e.g., CGSV), Aequa effectively addresses the issue of free-riders. In this scenario, the model width allocated to each participant is determined by their assessed contribution. Consequently, clients with lower contributions receive narrower model widths and thus lack access to the full-width model. This mechanism naturally restricts the benefits available to potential free riders and removes the explicit need for TEE.
>
> **Q3:** Our approach trains only two width configurations simultaneously in each forward pass. As demonstrated above, this approach does not introduce significant overhead. Specifically, we conducted an experiment on the FEMNIST dataset with a high number of clients (**3597 clients**), as detailed above and in the response to [Reviewer wyh2, W3]. The empirical results confirm that the overhead remains notably lower than $2\times$ (1.309) since our training leverages subnetworks. Thus, our method remains computationally efficient and scalable even with large models or substantial client counts.

---

### Decision · Program_Chairs · 2025-05-01

**Decision:**

Accept (poster)

**Comment:**

This paper introduces Aequa, a framework for collaborative federated learning to improve reward fairness with respect to their contributions. The key technique is repurposing slimmable networks as the media for reward allocation. The paper conducts a comprehensive theoretical analysis of the reward allocation mechanism and empirically validates it on six diverse datasets. All reviewers are satisfied and lean towards acceptance after reading the author's response. There are still some intrinsic limitations of Aequa, e.g., requiring the TEE environment to mitigate the free-riding problems. Overall, it is a solid contribution to fair collaborative learning. Authors should incorporate the feedback in the camera-ready version, e.g., adding the FEMNIST scalability experiment.